# A necessary and sufficient stability notion for adaptive generalization

**Katrina Ligett**
School of Computer Science & Engineering
Hebrew University of Jerusalem
Jerusalem 91904, Israel
katrina@cs.huji.ac.il

**Moshe Shenfeld**
School of Computer Science & Engineering
Hebrew University of Jerusalem
Jerusalem 91904, Israel
moshe.shenfeld@cs.huji.ac.il

## Abstract

We introduce a new notion of the stability of computations, which holds under post-processing and adaptive composition. We show that the notion is both necessary and sufficient to ensure generalization in the face of adaptivity, for any computations that respond to bounded-sensitivity linear queries while providing accuracy with respect to the data sample set. The stability notion is based on quantifying the effect of observing a computation's outputs on the posterior over the data sample elements. We show a separation between this stability notion and previously studied notion and observe that all differentially private algorithms also satisfy this notion.

## 1 Introduction

A fundamental idea behind most forms of data-driven research and machine learning is the concept of *generalization*–the ability to infer properties of a data distribution by working only with a sample from that distribution. One typical approach is to invoke a concentration bound to ensure that, for a sufficiently large sample size, the evaluation of the function on the sample set will yield a result that is close to its value on the underlying distribution, with high probability. Intuitively, these concentration arguments ensure that, for any given function, most sample sets are good "representatives" of the distribution. Invoking a union bound, such a guarantee easily extends to the evaluation of multiple functions on the same sample set.

Of course, such guarantees hold only if the functions to be evaluated were chosen independently of the sample set. In recent years, grave concern has erupted in many data-driven fields, that *adaptive selection* of computations is eroding statistical validity of scientific findings [Ioa05, GL14]. Adaptivity is not an evil to be avoided—it constitutes a natural part of the scientific process, wherein previous findings are used to develop and refine future hypotheses. However, unchecked adaptivity can (and does, as demonstrated by, e.g., [DFH+15b] and [RZ16]) often lead one to evaluate *overfitting* functions—ones that return very different values on the sample set than on the distribution.

Traditional generalization guarantees do not necessarily guard against adaptivity; while generalization ensures that the response to a query on a sample set will be *close* to that of the same query on the distribution, it does not rule out the possibility that the probability to get a *specific* response will be dramatically affected by the contents of the sample set. In the extreme, a generalizing computation could encode the whole sample set in the low-order bits of the output, while maintaining high accuracy with respect to the underlying distribution. Subsequent adaptive queries could then, by *post-processing* the computation's output, arbitrarily overfit to the sample set.

In recent years, an exciting line of work, starting with Dwork et al. [DFH+15b], has formalized this problem of adaptive data analysis and introduced new techniques to ensure guarantees of generalization in the face of an adaptively-chosen sequence of computations (what we call here

*adaptive generalization*). One great insight of Dwork et al. and followup work was that techniques for ensuring the *stability* of computations (some of them originally conceived as privacy notions) can be powerful tools for providing adaptive generalization.

A number of papers have considered variants of stability notions, the relationships between them, and their properties, including generalization properties. Despite much progress in this space, one issue that has remained open is the limits of stability—how much can the stability notions be relaxed, and still imply generalization? It is this question that we address in this paper.

## 1.1 Our Contribution

We introduce a new notion of the stability of computations, which holds under post-processing (Theorem 2.3) and adaptive composition (Theorems 2.6 and 2.7), and show that the notion is both necessary (Theorem 3.6) and sufficient (Theorem 3.3) to ensure generalization in the face of adaptivity, for any computations that respond to bounded-sensitivity linear queries (see Definition 3.1) while providing accuracy with respect to the data sample set. This means (up to a small caveat)[1] that our stability definition is equivalent to generalization, assuming sample accuracy, for bounded linear queries. Linear queries form the basis for many learning algorithms, such as those that rely on gradients or on the estimation of the average loss of a hypothesis.

In order to formulate our stability notion, we consider a prior distribution over the database elements and the posterior distribution over those elements conditioned on the output of a computation. In some sense, harmful outputs are those that induce large statistical distance between this prior and posterior (Definition 2.1). Our new notion of stability, *Local Statistical Stability* (Definition 2.2), intuitively, requires a computation to have only small probability of producing such a harmful output.

In Section 4, we directly prove that Differential Privacy, Max Information, Typical Stability and Compression Schemes all imply Local Statistical Stability, which provides an alternative method to establish their generalization properties. We also provide a few separation results between the various definitions.

## 1.2 Additional Related Work

Most countermeasures to overfitting fall into one of a few categories. A long line of work bases generalization guarantees on some form of bound on the complexity of the range of the mechanism, e.g., its VC dimension (see [SSBD14] for a textbook summary of these techniques). Other examples include *Bounded Description Length* [DFH+15a], and *compression schemes* [LW86] (which additionally hold under post-processing and adaptive composition [DFH+15a, CLN+16]). Another line of work focuses on the algorithmic stability of the computation [BE02], which bounds the effects on the output of changing one element in the training set.

A different category of stability notions, which focus on the effect of a small change in the sample set on the probability distribution over the range of possible outputs, has recently emerged from the notion of Differential Privacy [DMNS06]. Work of [DFH+15b] established that Differential Privacy, interpreted as a stability notion, ensures generalization; it is also known (see [DR+14]) to be robust to adaptivity and to withstand post-processing. A number of subsequent works propose alternative stability notions that weaken the conditions of Differential Privacy in various ways while attempting to retain its desirable generalization properties. One example is *Max Information* [DFH+15a], which shares the guarantees of Differential Privacy. A variety of other stability notions ([RRST16, RZ16, RRT+16, BNS+16, FS17, EGI19]), unlike Differential Privacy and Max Information, only imply generalization in expectation. [XR17, Ala17, BMN+17] extend these guarantees to generalization in probability, under various restrictions.

[CLN+16] introduce the notion of *post-hoc generalization*, which captures robustness to post-processing, but it was recently shown not to hold under composition [NSS+18]. The challenges that the internal correlation of non-product distributions present for stability have been studied in the context of *Inferential Privacy* [GK16] and *Typical Stability* [BF16].

## 2 LS stability definition and properties

Let $\mathcal{X}$ be an arbitrary countable *domain*. Fixing some $n \in \mathbb{N}$, let $D_{\mathcal{X}^n}$ be some probability distribution defined over $\mathcal{X}^n$.[2] Let $\mathcal{Q}, \mathcal{R}$ be arbitrary countable sets which we will refer to as *queries* and *responses*, respectively. Let a *mechanism* $M : \mathcal{X}^n \times \mathcal{Q} \to \mathcal{R}$ be a (possibly non-deterministic) function that, given a *sample set* $s \in \mathcal{X}^n$ and a query $q \in \mathcal{Q}$, returns a response $r \in \mathcal{R}$. Intuitively, queries can be thought of as questions the mechanism is asked about the sample set, usually representing functions from $\mathcal{X}^n$ to $\mathcal{R}$; the mechanism can be thought of as providing an estimate to the value of those functions, but we do not restrict the definitions, for reasons which will become apparent once we introduce the notion of adaptivity (Definition 2.4).

This setting involves two sources of randomness, the *underlying distribution* $D_{\mathcal{X}^n}$, and the *conditional distribution* $D_{\mathcal{R}|\mathcal{X}^n}^q (r \,|\, s)$—that is, the probability to get $r$ as the output of $M(s, q)$. These in turn induce a set of distributions (formalized in Definition A.1): the *marginal distribution* over $\mathcal{R}$, the *joint distribution* (denoted $D_{(\mathcal{X}^n, \mathcal{R})}^q$) and *product distribution* (denoted $D_{\mathcal{X}^n \otimes \mathcal{R}}^q$) over $\mathcal{X}^n \times \mathcal{R}$, and the *conditional distribution* over $\mathcal{X}^n$ given $r \in \mathcal{R}$. Note that even if $D_{\mathcal{X}^n}$ is a product distribution, this conditional distribution might not be a product distribution. Although the underlying distribution $D_{\mathcal{X}^n}$ is defined over $\mathcal{X}^n$, it induces a natural probability distribution over $\mathcal{X}$ as well, by sampling one of the sample elements in the set uniformly at random.[3] This in turn allows us extend our definitions to several other distributions, which form a connection between $\mathcal{R}$ and $\mathcal{X}$ (formalized in Definition A.2): the *marginal distribution* over $\mathcal{X}$, the *joint distribution* and *product distribution* over $\mathcal{X} \times \mathcal{R}$, the *conditional distribution* over $\mathcal{R}$ given $x \in \mathcal{X}$, and the *conditional distribution* over $\mathcal{X}$ given $r \in \mathcal{R}$. We use our distribution notation to denote both the probability that a distribution places on a subset of its range and the probability placed on a single element of the range.

**Notational conventions** We use calligraphic letters to denote domains, lower case letters to denote elements of these domains, capital letters to denote random variables taking values in these domains, and bold letters to denote subsets of these domains. We omit subscripts and superscripts from some notation when they are clear from context.

### 2.1 Local Statistical Stability

Before observing any output from the mechanism, an outside observer knowing $D$ but without other information about the sample set $s$ holds prior $D(x)$ that sampling an element of $s$ would return a particular $x \in \mathcal{X}$. Once an output $r$ of the mechanism is observed, however, the observer's posterior becomes $D(x \,|\, r)$. The difference between these two distributions is what determines the resulting degradation in stability. This difference could be quantified using a variety of distance measures (a partial list can be found in Appendix F); here we introduce a particular one which we use to define our stability notion.

**Definition 2.1** (Stability loss of a response). Given a distribution $D_{\mathcal{X}^n}$, a query $q$, and a mechanism $M : \mathcal{X}^n \times \mathcal{Q} \to \mathcal{R}$, the *stability loss* $\ell_{D_{\mathcal{X}^n}}^q (r)$ of a response $r \in \mathcal{R}$ with respect to $D_{\mathcal{X}^n}$ and $q$ is defined as the Statistical Distance (Definition F.1) between the prior distribution over $\mathcal{X}$ and the posterior induced by $r$. That is,

$$\ell_{D_{\mathcal{X}^n}}^q (r) := \sum_{x \in \mathbf{x}_+(r)} \left( D(x \,|\, r) - D(x) \right),$$

where $\mathbf{x}_+(r) := \{x \in \mathcal{X} \,|\, D(x \,|\, r) > D(x)\}$, the set of all sample elements which have a posterior probability (given $r$) higher then their prior. Similarly, we define the stability loss $\ell(\mathbf{r})$ of a set of responses $\mathbf{r} \subseteq \mathcal{R}$ as

$$\ell(\mathbf{r}) := \frac{\sum_{r \in \mathbf{r}} D(r) \cdot \ell(r)}{D(\mathbf{r})}.$$

Given $0 \leq \epsilon \leq 1$, a response will be called $\epsilon$-*unstable* with respect to $D_{\mathcal{X}^n}$ and $q$ if its loss is greater the $\epsilon$. The set of all $\epsilon$-unstable responses will be denoted $\mathbf{r}_\epsilon^{D_{\mathcal{X}^n}, q} := \{r \in \mathcal{R} \,|\, \ell(r) > \epsilon\}$.

We now introduce our notion of stability of a mechanism.

**Definition 2.2** (Local Statistical Stability). Given $0 \leq \epsilon, \delta \leq 1$, a distribution $D_{\mathcal{X}^n}$, and a query $q$, a mechanism $M : \mathcal{X}^n \times \mathcal{Q} \to \mathcal{R}$ will be called $(\epsilon, \delta)$-*Local-Statistically Stable with respect to* $D_{\mathcal{X}^n}$ *and q* (or *LS Stable*, or *LSS*, for short) if for any $\mathbf{r} \subseteq \mathcal{R}$, $D(\mathbf{r}) \cdot (\ell(\mathbf{r}) - \epsilon) \leq \delta$.

Notice that the maximal value of the left hand side is achieved for the subset $\mathbf{r}_\epsilon$. This stability definition can be extended to apply to a family of queries and/or a family of possible distributions. When there exists a family of queries $\mathcal{Q}$ and a family of distributions $\mathcal{D}$ such that a mechanism $M$ is $(\epsilon, \delta)$-LSS for all $D_{\mathcal{X}^n} \in \mathcal{D}$ and for all $q \in \mathcal{Q}$, then $M$ will be called $(\epsilon, \delta)$-*LSS for* $\mathcal{D}, \mathcal{Q}$. (This stability notion somewhat resembles *Semantic Privacy* as discussed by [KS14], though they use it to compare different posterior distributions.)

Intuitively, this can be thought of as placing a $\delta$ bound on the probability of observing an outcome whose stability loss exceeds $\epsilon$. This claim is formalized in Lemma B.1.

## 2.2 Properties

We now turn to prove two crucial properties of LSS: post-processing and adaptive composition.

Post-processing guarantees (known in some contexts as data processing inequalities) ensure that the stability of a computation can only be *increased* by subsequent manipulations. This is a key desideratum for concepts used to ensure adaptivity-proof generalization, since otherwise an adaptive subsequent computation could potentially arbitrarily degrade the generalization guarantees.

**Theorem 2.3** (LSS holds under Post-Processing). *Given* $0 \leq \epsilon, \delta \leq 1$, *a distribution* $D_{\mathcal{X}^n}$, *and a query q, if a mechanism $M$ is $(\epsilon, \delta)$-LSS with respect to $D_{\mathcal{X}^n}$ and q, then for any range $\mathcal{U}$ and any arbitrary (possibly non-deterministic) function $f : \mathcal{R} \to \mathcal{U}$, we have that $f \circ M : \mathcal{X}^n \times \mathcal{Q} \to \mathcal{U}$ is also $(\epsilon, \delta)$-LSS with respect to $D_{\mathcal{X}^n}$ and q. An analogous statement also holds for mechanisms that are LSS with respect to a family of queries and/or a family of distributions.*

The proof appears in Appendix B.1.

In order to formally define adaptive learning and stability under adaptively chosen queries, we formalize the notion of an analyst who issues those queries.

**Definition 2.4** (Analyst and Adaptive Mechanism). An *analyst over a family of queries* $\mathcal{Q}$ is a (possibly non-deterministic) function $A : \mathcal{R}^* \to \mathcal{Q}$ that receives a *view*—a finite sequence of responses—and outputs a query. We denote by $\mathcal{A}$ the family of all analysts, and write $\mathcal{V}_k := \mathcal{R}^k$ and $\mathcal{V} := \mathcal{R}^*$.

Illustrated below, the *adaptive mechanism* $\text{Adp}_{\bar{M}} : \mathcal{X}^n \times \mathcal{A} \to \mathcal{V}_k$ is a particular type of mechanism, which inputs an analyst as its query and which returns a view as its range type. It is parameterized by a sequence of *sub-mechanisms* $\bar{M} = (M_i)_{i=1}^k$ where $\forall i \in [k]$, $M_i : \mathcal{X}^n \times \mathcal{Q} \to \mathcal{R}$. Given a sample set $s$ and an analyst $A$ as input, the adaptive mechanism iterates $k$ times through the process where $A$ sends a query to $M_i$ and receives its response to that query on the sample set. The adaptive mechanism returns the resulting sequence of $k$ responses $v_k$. Naturally, this requires $A$ to match $M$ such that $M$'s range can be $A$'s input, and vice versa.[4][5]

For illustration, consider a gradient descent algorithm, where at each step the algorithm requests an estimate of the gradient at a given point, and chooses the next point in which the gradient should be evaluated based on the response it receives. For us, $M$ evaluates the gradient at a given point, and $A$

```
Adaptive Mechanism Adp_M̄
─────────────────────────────────
Input: s ∈ 𝒳ⁿ, A ∈ 𝒜
Output: v_k ∈ 𝒱_k
v₀ ← ∅ or c
for i ∈ [k] :
    q_i ← A(v_{i-1})
    r_i ← M_i(s, q_i)
    v_i ← (v_{i-1}, r_i)
return v_k
```

determines the next point to be considered. The interaction between the two of them constitutes an adaptive learning process.

**Definition 2.5** ($k$-LSS under adaptivity). Given $0 \leq \epsilon, \delta \leq 1$, a distribution $D_{\mathcal{X}^n}$, and an analyst $A$, a sequence of $k$ mechanisms $\bar{M}$ will be called $(\epsilon, \delta)$-*local-statistically stable under $k$ adaptive iterations* with respect to $D_{\mathcal{X}^n}$ and $A$ (or $k$-LSS for short), if $\text{Adp}_{\bar{M}}$ is $(\epsilon, \delta)$-LSS with respect to $D_{\mathcal{X}^n}$ and $A$ (in which case we will use $\mathbf{v}_{\epsilon}^{k, A, D_{\mathcal{X}^n}}$ to denote the set of $\epsilon$ unstable views). This definition can be extended to a family of analysts and/or a family of possible distributions as well.

Adaptive composition is a key property of a stability notion, since it restricts the degradation of stability across multiple computations. A key observation is that the posterior $D(s \mid v_k)$ is itself a distribution over $\mathcal{X}^n$ and $q_{k+1}$ is a deterministic function of $v_k$. Therefore, as long as each sub-mechanism is LSS with respect to any posterior that could have been induced by previous adaptive interaction, one can reason about the properties of the composition.

We first show that the stability loss of a view is bounded by the sum of losses of its responses with respect to the sub-mechanisms, which provides a linear bound on the degradation of the LSS parameters. Adding a bound on the expectation of the loss of the sub-mechanisms allows us to also invoke Azuma's inequality and prove a sub-linear bound.

**Theorem 2.6** (LSS adaptively composes linearly). *Given a family of distributions $\mathcal{D}$ over $\mathcal{X}^n$, a family of queries $\mathcal{Q}$, and a sequence of $k$ mechanisms $\bar{M}$ where $\forall i \in [k]$, $M_i : \mathcal{X}^n \times \mathcal{Q} \to \mathcal{R}$, we will denote $\mathcal{D}_{M_0, \mathcal{Q}} := \mathcal{D}$, and for any $i > 0$, $\mathcal{D}_{M_i, \mathcal{Q}}$ will denote the set of all posterior distributions induced by any response of $M_i$ with non-zero probability with respect to $\mathcal{D}_{M_{i-1}, \mathcal{Q}}$ and $\mathcal{Q}$ (see Definition B.2).*

*Given a sequence $0 \leq \epsilon_1, \delta_1, \ldots, \epsilon_k, \delta_k \leq 1$, if for all $i$, $M_i$ is $(\epsilon_i, \delta_i)$-LSS with respect to $\mathcal{D}_{M_{i-1}, \mathcal{Q}}$ and $\mathcal{Q}$, the sequence is $\left( \sum_{i \in [k]} \epsilon_i, \sum_{i \in [k]} \delta_i \right)$-$k$-LSS with respect to $\mathcal{D}$ and any analyst $A$ over $\mathcal{Q} \times \mathcal{R}$.*

The proof appears in Appendix B.3.

One simple case is when $\mathcal{D}_{M_{i-1}, \mathcal{Q}} = \mathcal{D}$, and $M_i$ is $(\epsilon_i, \delta_i)$-LSS with respect to $\mathcal{D}$ and $\mathcal{Q}$, for all $i$.

**Theorem 2.7** (LSS adaptively composes sub-linearly). *Under the same conditions as Theorem 2.6, given $0 \leq \alpha_1, \ldots, \alpha_k \leq 1$, such that for all $i$ and any $D_{\mathcal{X}^n} \in \mathcal{D}_{M_{i-1}, \mathcal{Q}}$, and $q \in \mathcal{Q}$,*

$$\mathbb{E}_{S \sim D_{\mathcal{X}^n}, R \sim M_i(s,q)} [\ell(R)] \leq \alpha_i, \text{ then for any } 0 \leq \delta' \leq 1, \text{ the sequence is } \left( \epsilon', \delta' + \sum_{i \in [k]} \frac{\delta_i}{\epsilon_i} \right) \text{-}k\text{-}$$

*LSS with respect to $\mathcal{D}$ and any analyst $A$ over $\mathcal{Q} \times \mathcal{R}$, where $\epsilon' := \sqrt{8 \ln\left(\frac{1}{\delta'}\right) \sum_{i \in [k]} \epsilon_i^2} + \sum_{i \in [k]} \alpha_i$.*

The theorem provides a better bound then the previous one in case $\alpha_i \ll \epsilon_i$, in which case the dominating term is the first one, which is sub-linear in $k$. The proof appears in Appendix B.4.

## 3 LSS is Necessary and Sufficient for Generalization

Up until this point, queries and responses have been fairly abstract concepts. In order to discuss generalization and accuracy, we must make them concrete. As a result, in this section, we often consider queries in the family of functions $q : \mathcal{X}^n \to \mathcal{R}$, and consider responses which have some

metric defined over them. We show our results for a fairly general class of functions known as bounded linear queries.[6]

**Definition 3.1** (Linear queries). A function $q : \mathcal{X}^n \to \mathbb{R}$ will be called a *linear query*, if it is defined by a function $q_1 : \mathcal{X} \to \mathbb{R}$ such that $q(s) := \frac{1}{n} \sum_{i=1}^{n} q_1(s_i)$ (for simplicity we will denote $q_1$ simply as $q$ throughout the paper). If $q : \mathcal{X} \to [-\Delta, \Delta]$ it will be called a $\Delta$-*bounded linear query*. The set of $\Delta$-bounded linear queries will be denoted $\mathcal{Q}_\Delta$.

In this context, there is a "correct" answer the mechanism can produce for a given query, defined as the value of the function on the sample set or distribution, and its distance from the response provided by the mechanism can be thought of as the mechanism's error.

**Definition 3.2** (Sample accuracy, distribution accuracy). Given $0 \le \epsilon$, $0 \le \delta \le 1$, a distribution $D_{\mathcal{X}^n}$, and a query $q$, a mechanism $M : \mathcal{X}^n \times \mathcal{Q} \to \mathbb{R}$ will be called $(\epsilon, \delta)$-*Sample Accurate with respect to $D_{\mathcal{X}^n}$ and $q$*, if

$$\Pr_{S \sim D_{\mathcal{X}^n}, R \sim M(S,q)} [|R - q(S)| > \epsilon] \le \delta.$$

Such a mechanism will be called $(\epsilon, \delta)$-*Distribution Accurate with respect to $D_{\mathcal{X}^n}$ and $q$* if

$$\Pr_{S \sim D_{\mathcal{X}^n}, R \sim M(S,q)} [|R - q(D_{\mathcal{X}^n})| > \epsilon] \le \delta,$$

where $q(D_{\mathcal{X}^n}) := \mathbb{E}_{S \sim D_{\mathcal{X}^n}} [q(S)]$. When there exists a family of distributions $\mathcal{D}$ and a family of queries $\mathcal{Q}$ such that a mechanism $M$ is $(\epsilon, \delta)$-Sample (Distribution) Accurate for all $D \in \mathcal{D}$ and for all $q \in \mathcal{Q}$, then $M$ will be called $(\epsilon, \delta)$-*Sample (Distribution) Accurate with respect to $\mathcal{D}$ and $\mathcal{Q}$*.

A sequence of $k$ mechanisms $\bar{M}$ where $\forall i \in [k] : M_i : \mathcal{X}^n \times \mathcal{Q} \to \mathbb{R}$ which respond to a sequence of $k$ (potentially adaptively chosen) queries $q_1, \ldots q_k$ will be called $(\epsilon, \delta)$-$k$-*Sample Accurate with respect to $D_{\mathcal{X}^n}$ and $q_1, \ldots q_k$* if

$$\Pr_{S \sim D_{\mathcal{X}^n}, R_i \sim M_i(S,q_i)} \left[ \max_{i \in k} |R_i - q_i(S)| > \epsilon \right] \le \delta,$$

and $(\epsilon, \delta)$-$k$-*Distribution Accurate with respect to $D_{\mathcal{X}^n}$ and $q_1, \ldots q_k$* if

$$\Pr_{S \sim D_{\mathcal{X}^n}, R_i \sim M_i(S,q_i)} \left[ \max_{i \in k} |R_i - q_i(D_{\mathcal{X}^n})| > \epsilon \right] \le \delta.$$

When considering an adaptive process, accuracy is defined with respect to the analyst, and the probabilities are taken also over the choice of the coin tosses by the adaptive mechanism.[7]

We denote by $\mathbb{V}$ the set of views consisting of responses in $\mathbb{R}$.

We now show that if a mechanism returns accurate results with respect to the sample set, then being LSS implies accuracy on the underlying distribution.

**Theorem 3.3** (LSS implies generalization with high probability). *Given $0 \le \epsilon \le \Delta$, $0 \le \delta \le 1$, a distribution $D_{\mathcal{X}^n}$, and an analyst $A : \mathbb{V} \to \mathcal{Q}_\Delta$, if a sequence of $k$ mechanisms $\bar{M}$ where $\forall i \in [k], M_i : \mathcal{X}^n \times \mathcal{Q}_\Delta \to \mathbb{R}$ is both $\left( \frac{\epsilon}{8\Delta}, \frac{\epsilon^2 \delta}{4800\Delta^2} \right)$-$k$-LSS and $\left( \frac{\epsilon}{8}, \frac{\epsilon \delta}{600\Delta} \right)$-$k$-Sample Accurate with respect to $D_{\mathcal{X}^n}$ and $A$, then it is $(\epsilon, \delta)$-$k$-Distribution Accurate with respect to $D_{\mathcal{X}^n}$ and $A$.*

The proof of this theorem consists of two stages, and follows the method introduced by [BNS+16]. First we show that the a query returned by an LSS mechanism has expected value on the underlying distribution that is close to its value on the sample set that the mechanism received as input (Appendix C.1). We then proceed to lift this guarantee from expectation to high probability, using a thought experiment known as the *Monitor Mechanism* (Appendix C.2). Intuitively, it runs a large number of

independent copies of an underlying mechanism, and exposes the results of the least-distribution-accurate copy as its output. If the expected error of even this least-accurate-copy is relatively low, then the underlying mechanism generalizes with high probability (Appendix C.3).

We next show that a mechanism that is not LSS cannot be both Sample Accurate and Distribution Accurate. In order to prove this theorem, we show how to construct a "bad" query.

**Definition 3.4** (Loss assessment query). Given a query $q$ and a response $r$, we will define the *Loss assessment query $\tilde{q}_r$* as

$$\tilde{q}_r(x) = \begin{cases} \Delta & D(x) > D(x \mid r) \\ -\Delta & D(x) \leq D(x \mid r) \end{cases}.$$

Intuitively, this function maximizes the difference between $\underset{X \sim D_{\mathcal{X}}}{\mathbb{E}}[\tilde{q}_r(X)]$ and $\underset{X \sim D_{\mathcal{X} \mid \mathcal{R}}^q}{\mathbb{E}}[\tilde{q}_r(X) \mid r]$, and as a result, the potential to overfit.[8]

This function is used to lower bound the effect of the stability loss on the expected overfitting.

**Lemma 3.5** (Loss assessment query overfits in expectation). *Given $0 \leq \epsilon, \delta \leq 1$, a distribution $D_{\mathcal{X}^n}$, a query $q$, and a mechanism $M$, if $D(\mathbf{r}_\epsilon) > \delta$, then there is a function $f : \mathcal{R} \to \mathcal{Q}_\Delta$ such that,*

$$\left| \underset{S \sim D_{\mathcal{X}^n}, Q' \sim f \circ M(S,q)}{\mathbb{E}}[Q'(D_{\mathcal{X}^n}) - Q'(S)] \right| > 2\epsilon \Delta \delta.$$

*Proof.* Choosing $f(r) = q_r$ we get that,

$$\left| \underset{S \sim D_{\mathcal{X}^n}, Q' \sim f \circ M(S,q)}{\mathbb{E}}[Q'(D_{\mathcal{X}^n}) - Q'(S)] \right| \overset{(1)}{=} \left| \sum_{q' \in \mathcal{Q}_\Delta} D(q') \cdot \sum_{x \in \mathcal{X}} (D(x) - D(x \mid q')) \cdot q'(x) \right|$$

$$= \left| \sum_{r \in \mathcal{R}} D(r) \cdot \sum_{x \in \mathcal{X}} (D(x) - D(x \mid r)) \cdot \tilde{q}_r(x) \right|$$

$$\overset{(2)}{\geq} \overbrace{\sum_{r \in \mathbf{r}_\epsilon} D(r)}^{\geq \delta} \cdot \overbrace{\sum_{x \in \mathcal{X}} |D(x) - D(x \mid r)| \cdot \Delta}^{=2\ell(r) > 2\epsilon}$$

$$\overset{(3)}{>} 2\epsilon \Delta \delta$$

where (1) is further justified in the proof of Theorem C.1, (2) results from the definition of the loss assessment query, and (3) from the definition of $\mathbf{r}_\epsilon$. $\square$

We use this method for constructing an overfitting query for non-LSS mechanism, to show that LSS is necessary in order for a mechanism to be both Sample Accurate and Distribution Accurate.

**Theorem 3.6** (Necessity of LSS for Generalization). *Given $0 \leq \epsilon \leq \Delta$, $0 \leq \delta \leq 1$, a distribution $D_{\mathcal{X}^n}$, and an analyst $A : \mathbb{V} \to \mathcal{Q}_\Delta$, if a sequence of $k$ mechanisms $\bar{M}$ where $\forall i \in [k], M_i : \mathcal{X}^n \times \mathcal{Q}_\Delta \to \mathbb{R}$ is not $\left(\frac{\epsilon}{\Delta}, \delta\right)$-$k$-LSS, then it cannot be both $\left(\frac{\epsilon}{5}, \frac{\epsilon\delta}{5\Delta}\right)(k+1)$-Distribution Accurate and $\left(\frac{\epsilon}{5}, \frac{\epsilon\delta}{5\Delta}\right)(k+1)$-Sample Accurate.*

The proof of this theorem, which appears in Appendix C.4, uses a similar method to the proof of Theorem 3.3, employing a variant of the Monitor Mechanism that outputs the loss assessment query with the highest level of overfitting.

## 4 Relationship to other notions of stability

In this section, we discuss the relationship between LSS and a few common notions of stability; definitions can be found in Appendix D.1. In order to do so, we introduce an additional new stability notion, which relaxes the Max Information (MI) (Definition D.2) notion by moving from the distribution over the sample sets to the distribution over the sample elements.

**Definition 4.1** (Local Max Information). Given $0 \leq \epsilon$, $0 \leq \delta \leq 1$, a distribution $D_{\mathcal{X}^n}$ and a query $q$, a mechanism $M$ will be said to satisfy $(\epsilon, \delta)$-*Local-Max-Information with respect to $D_{\mathcal{X}^n}$ and $q$* (or *LMI*, for short), if the joint distributions $D_{(\mathcal{X}, \mathcal{R})}$ and the product distribution $D_{\mathcal{X} \otimes \mathcal{R}}$ over $\mathcal{X} \times \mathcal{R}$ are $(\epsilon, \delta)$-indistinguishable. In other words, for any $\mathbf{b} \subseteq \mathcal{X} \times \mathcal{R}$,

$$D_{(\mathcal{X}, \mathcal{R})}(\mathbf{b}) \leq e^\epsilon \cdot D_{\mathcal{X} \otimes \mathcal{R}}(\mathbf{b}) + \delta \quad \text{and} \quad D_{\mathcal{X} \otimes \mathcal{R}}(\mathbf{b}) \leq e^\epsilon \cdot D_{(\mathcal{X}, \mathcal{R})}(\mathbf{b}) + \delta.$$

The definition can be extended to apply to a family of queries and/or a family of possible distributions.

## 4.1 Implications

Prior work ([DFH⁺15a] and [RRST16]) showed that bounded Differential Privacy (DP) (Definition D.1) implies bounded MI (Definition D.2). In the case of $\delta > 0$, this holds only if the underlying distribution is a product distribution [De12]). Bounded MI is also implied by Typical Stability (TS) (Definition D.3) [BF16], and Bounded Maximal Leakage (ML) [EGI19]. We prove that DP, MI and TS imply LMI (in the case of DP, only for product distributions). All proofs for this subsection can be found in Appendix D.2, where we also introduce a local version of ML and prove its relation to LMI.

**Theorem 4.2** (Differential Privacy implies Local Max Information). *Given $0 \leq \epsilon$, $0 \leq \delta \leq 1$, a distribution $D_{\mathcal{X}}$, and a query $q$, if a mechanism $M$ is $(\epsilon, \delta)$-DP with respect to $q$ then it is $(\epsilon, \delta)$-LMI with respect to the same $q$ and the product distribution over $\mathcal{X}^n$ induced by $D_{\mathcal{X}}$.*

**Theorem 4.3** (Max Information implies Local Max Information). *Given $0 \leq \epsilon$, $0 \leq \delta \leq 1$, a distribution $D_{\mathcal{X}^n}$ and a query $q$, if a mechanism $M$ has $\delta$-approximate max-information of $\epsilon$ with respect to $D_{\mathcal{X}^n}$ and $q$ then it is $(\epsilon, \delta)$-LMI with respect to the same $D_{\mathcal{X}^n}$ and $q$.*

**Theorem 4.4** (Typical Stability implies Local Max Information). *Given $0 \leq \epsilon$, $0 \leq \delta, \eta \leq 1$, a distribution $D_{\mathcal{X}^n}$ and a query $q$, if a mechanism $M$ is $(\epsilon, \delta, \eta)$-Typically Stable with respect to $D_{\mathcal{X}^n}$ and $q$ then it is $(\epsilon, \delta + 2\eta)$-LMI with respect to the same $D_{\mathcal{X}^n}$ and $q$.*

These three theorems follow naturally from the fact that LMI is a fairly direct relaxation of DP, MI and TS.

We next show that LMI implies LSS.

**Theorem 4.5** (Local Max Information implies Local Statistical Stability). *Given $0 \leq \delta \leq \epsilon \leq \frac{1}{3}$, a distribution $D_{\mathcal{X}^n}$ and a query $q$, if a mechanism $M$ is $(\epsilon, \delta)$-LMI with respect to $D_{\mathcal{X}^n}$ and $q$, then it is $\left(\epsilon', \frac{\delta}{\epsilon}\right)$-LSS with respect to the same $D_{\mathcal{X}^n}$ and $q$, where $\epsilon' = e^\epsilon - 1 + \epsilon$.*

We also prove that Compression Schemes (Definition D.6) imply LSS. This results from the fact that releasing information based on a restricted number of sample elements has a limited effect on the posterior distribution on one element of the sample set.

**Theorem 4.6** (Compressibility implies Local Statistical Stability). *Given $0 \leq \delta \leq 1$, an integer $m \leq \frac{n}{9\ln(2n/\delta)}$, a distribution $D_{\mathcal{X}}$, and a query $q \in \mathcal{Q}$, if a mechanism $M$ has a compression scheme of size $m$ then it is $(\epsilon, \delta)$-LSS with respect to the same $q$ and the product distribution over $\mathcal{X}^n$ induced by $D_{\mathcal{X}}$, for any $\epsilon > 11\sqrt{\frac{m\ln(2n/\delta)}{n}}$.[9]*

## 4.2 Separations

Finally, we show that MI is a strictly stronger requirement than LMI, and LMI is a strictly stronger requirement then LSS. Proofs of these theorems appear in Appendix D.3.

**Theorem 4.7** (Max Information is strictly stronger than Local Max Information). *For any $0 < \epsilon$, $n \geq 3$, the mechanism which outputs the parity function of the sample set is $(\epsilon, 0)$-LMI but not $\left(1, \frac{1}{5}\right)$-MI.*

**Theorem 4.8** (Local Max Information is strictly stronger than Local Statistical Stability). *For any $0 \leq \delta \leq 1$, $n > \max\left\{2\ln\left(\frac{2}{\delta}\right), 6\right\}$, a mechanism which uniformly samples and outputs one sample element is $\left(11\sqrt{\frac{\ln(2n/\delta)}{n}}, \delta\right)$-LSS but is not $\left(1, \frac{1}{2n}\right)$-LMI.*

# 5  Applications and Discussion

In order to make the LSS notion useful, we must identify mechanisms which manages to remain stable while maintaining sample accuracy. Fortunately, many such mechanisms have been introduced in the context of Differential Privacy. Two of the most basic Differentially Private mechanisms are based on noise addition, of either a Laplace or a Gaussian random variable. Careful tailoring of their parameters allows "masking" the effect of changing one element, while maintaining a limited effect on the sample accuracy. By Theorems 4.2 and 4.5, these mechanisms are guaranteed to be LSS as well. The definitions and properties of these mechanisms can be found in Appendix E.

In moving away from the study of worst-case data sets (as is common in previous stability notions) to averaging over sample sets and over data elements of those sets, we hope that the Local Statistical Stability notion will enable new progress in the study of generalization under adaptive data analysis. This averaging, potentially leveraging a sort of "natural noise" from the data sampling process, may enable the development of new algorithms to preserve generalization, and may also support tighter bounds on the implications of existing algorithms. One possible way this might be achieved is by limiting the family of distributions and queries, such that the empirical mean of the query lies within some confidence interval around population mean, which would allow scaling the noise to the interval rather than the full range (see, e.g. , *Concentrated Queries*, as proposed by [BF16]).

One might also hope that realistic adaptive learning settings are not adversarial, and might therefore enjoy even better generalization guarantees. LSS may be a tool for understanding the generalization properties of algorithms of interest (as opposed to worst-case queries or analysts; see e.g. [GK16], [ZH19]).

**Acknowledgements**   This work was supported in part by Israel Science Foundation (ISF) grant 1044/16, the United States Air Force and DARPA under contract FA8750-16-C-0022, and the Federmann Cyber Security Center in conjunction with the Israel national cyber directorate. Part of this work was done while the authors were visiting the Simons Institute for the Theory of Computing. Any opinions, findings and conclusions or recommendations expressed in this material are those of the authors and do not necessarily reflect the views of the United States Air Force and DARPA.

## Footnotes

[1]In particular, our lower bound (Theorem 3.6) requires one more query than our upper bound (Theorem 3.3).

[2]Throughout the paper, $\mathcal{X}^n$ can either denote the family of sequences of length $n$ or a multiset of size $n$; that is, the sample set $s$ can be treated as an ordered or unordered set.

[3]It is worth noting that in the case where $D_{\mathcal{X}^n}$ is the product distribution of some distribution $P_{\mathcal{X}}$ over $\mathcal{X}$, we get that the induced distribution over $\mathcal{X}$ is $P_{\mathcal{X}}$.

[4]If the same mechanism appears more then once in $\bar{M}$, it can also be stateful, which means it retains an internal record consisting of internal randomness, the history of sample sets and queries it has been fed, and the responses it has produced; its behavior may be a function of this internal record. We omit this from the notation for simplicity, but do refer to this when relevant. A stateful mechanism will be defined as LSS if it is LSS given any reachable internal record. A pedantic treatment might consider the *probability* that a particular internal state could be reached, and only require LSS when accounting for these probabilities.

[5]If $A$ is randomized, we add one more step at the beginning where $\text{Adp}_{\bar{M}}$ randomly generates some bits $c$—$A$'s "coin tosses." In this case, $v_k := (c, r_1, \ldots, r_{ik})$ and $A$ receives the coin tosses as an input as well. This addition turns $q_{k+1}$ into a deterministic function of $v_i$ for any $i \in \mathbb{N}$, a fact that will be used multiple times throughout the paper. In this situation, the randomness of $\text{Adp}_{\bar{M}}$ results both from the randomness of the coin tosses and from that of the sub-mechanisms.

[6]For simplicity, throughout the following section we choose $\mathcal{R} = \mathbb{R}$, but all results extend to any metric space, in particular $\mathbb{R}^d$.

[7]If the adaptive mechanism invokes a stateful sub-mechanism multiple times, we specify that the mechanism is Sample (Distribution) Accurate if it is Sample (Distribution) Accurate given any reachable internal record. Again, a somewhat more involved treatment might consider the *probability* that a particular internal state of the mechanism could be reached.

[8]The fact that we are able to define such a query is a result of the way the distance measure of LSS treats the $x$'s and the fact that it is defined over $\mathcal{X}$ and not $\mathcal{X}^n$.

[9]In case $g$ releases some side information, the number of bits required to describe this information is added to the $m$ factor in the bound on $\epsilon$.

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
