[Supplementary Material · LSS - NeurIPS full - V2.2.pdf]

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

[10]If $M_{k+1}$ is stateful, the conditioning can result from any unknown state of $M_{k+1}$ which might affect its response to $q_{k+1}$. If $M_{k+1}$ has no shared state with the previous sub-mechanisms (either because it is a different mechanism or because it is stateless), then the only effect $v_k$ has on the posterior on $\mathcal{R}$ is by governing $q_{k+1}$ (which, as mentioned before, is a deterministic function of $v_k$ for the given $A$), in which case $P^{v_k}_{\mathcal{R}|\mathcal{X}^n}(r \mid s) = D^{q_{k+1}}_{\mathcal{R}|\mathcal{X}^n}(r \mid s)$ where the mechanism is $M_{k+1}$.

[11]If the analyst $A$ is non-deterministic, $\Omega_0:=\mathcal{X}^n\times C$, where $C$ is the set of all possible coin tosses of the analyst, as mentioned in Definition 2.4. If the mechanisms have some internal state not expressed by the responses, $\Omega_i$ will be the domain of those states, as mentioned in Definition B.2.

[12]Of course, no realistic mechanism would have such an ability; the monitor mechanism is simply a thought experiment used as a proof technique.

[13]The fact that repeating this process $t$ independent times affects only the $\delta$ and not the $\epsilon$ will be crucial to the move from generalization of expectation to generalization with high probability (at least in this proof technique). This is made possible by the way $r$'s were treated in the distance measure in the LSS definition. For comparison, see the remark in Lemma 3.3 in [BNS$^+$16]. We hypothesize, quite informally, that stability definitions that degrade in the $\epsilon$ term on multiple independent runs cannot yield generalization with high probability. As far as we are aware, all previously studied stability notions support this claim.

[14] some versions include the option of receiving some side information, i.e. the coin tosses of $g$.

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

# A  Distributions: Formal Definitions

**Definition A.1** (Distributions over $\mathcal{X}^n$ and $\mathcal{R}$)**.** A distribution $D_{\mathcal{X}^n}$, a query $q$, and a mechanism $M : \mathcal{X}^n \times \mathcal{Q} \to \mathcal{R}$, together induce a set of distributions over $\mathcal{X}^n$, $\mathcal{R}$, and $\mathcal{X}^n \times \mathcal{R}$.

The *conditional distribution* $D^q_{\mathcal{R}|\mathcal{X}^n}$ over $\mathcal{R}$ represents the probability to get $r$ as the output of $M(s, q)$. That is, $\forall s \in \mathcal{X}^n, r \in \mathcal{R}$,

$$D^q_{\mathcal{R}|\mathcal{X}^n}(r \mid s) := \Pr_{R \sim M(s,q)}[R = r \mid s],$$

where the probability is taken over the internal randomness of $M$.

The *joint distribution* $D^q_{(\mathcal{X}^n, \mathcal{R})}$ over $\mathcal{X}^n \times \mathcal{R}$ represents the probability to sample a particular $s$ and get $r$ as the output of $M(s, q)$. That is, $\forall s \in \mathcal{X}^n, r \in \mathcal{R}$,

$$D^q_{(\mathcal{X}^n, \mathcal{R})}(s, r) := D_{\mathcal{X}^n}(s) \cdot D^q_{\mathcal{R}|\mathcal{X}^n}(r \mid s).$$

The *marginal distribution* $D^q_{\mathcal{R}}$ over $\mathcal{R}$ represents the prior probability to get output $r$ without any knowledge of $s$. That is, $\forall r \in \mathcal{R}$,

$$D^q_{\mathcal{R}}(r) := \sum_{s \in \mathcal{X}^n} D^q_{(\mathcal{X}^n, \mathcal{R})}(s, r).$$

The *product distribution* $D^q_{\mathcal{X}^n \otimes \mathcal{R}}$ over $\mathcal{X}^n \times \mathcal{R}$ represents the probability to sample $s$ and get $r$ as the output of $M(\cdot, q)$ independently. That is, $\forall s \in \mathcal{X}, r \in \mathcal{R}$,

$$D^q_{\mathcal{X}^n \otimes \mathcal{R}}(s, r) := D_{\mathcal{X}^n}(s) \cdot D^q_{\mathcal{R}}(r).$$

The *conditional distribution* $D^q_{\mathcal{X}^n|\mathcal{R}}$ over $\mathcal{X}^n$ represents the posterior probability that the sample set was $s$ given that $M(\cdot, q)$ returns $r$. That is, $\forall s \in \mathcal{X}^n, r \in \mathcal{R}$,

$$D^q_{\mathcal{X}^n|\mathcal{R}}(s \mid r) := \frac{D^q_{(\mathcal{X}^n, \mathcal{R})}(s, r)}{D^q_{\mathcal{R}}(r)}.$$

**Definition A.2** (Distributions over $\mathcal{X}$ and $\mathcal{R}$)**.** The *marginal distribution* $D_{\mathcal{X}}$ over $\mathcal{X}$ represents the probability to get $x$ by sampling a sample set uniformly at random without any knowledge of $s$. That is, $\forall x \in \mathcal{X}$,

$$D_{\mathcal{X}}(x) := \sum_{s \in \mathcal{X}^n} D_{\mathcal{X}^n}(s) \cdot D_{\mathcal{X}|\mathcal{X}^n}(x \mid s),$$

where $D_{\mathcal{X}|\mathcal{X}^n}(x \mid s)$ denotes the probability to get $x$ by sampling $s$ uniformly at random.

The *joint distribution* $D^q_{(\mathcal{X}, \mathcal{R})}$ over $\mathcal{X} \times \mathcal{R}$ represents the probability to get $x$ by sampling a sample set uniformly at random and also get $r$ as the output of $M(\cdot, q)$ from the same sample set. That is, $\forall x \in \mathcal{X}, r \in \mathcal{R}$,

$$D^q_{(\mathcal{X}, \mathcal{R})}(x, r) := \sum_{s \in \mathcal{X}^n} D_{\mathcal{X}^n}(s) \cdot D_{\mathcal{X}|\mathcal{X}^n}(x \mid s) \cdot D^q_{\mathcal{R}|\mathcal{X}^n}(r \mid s).$$

where $D_{\mathcal{X}|\mathcal{X}^n}(x \mid s)$ denotes the probability to get $x$ by sampling $s$ uniformly at random.

The *product distribution* $D^q_{\mathcal{X} \otimes \mathcal{R}}$ over $\mathcal{X} \times \mathcal{R}$ represents the probability to get $x$ by sampling a sample set uniformly at random and get $r$ as the output of $M(\cdot, q)$ independently. That is, $\forall x \in \mathcal{X}, r \in \mathcal{R}$,

$$D^q_{\mathcal{X} \otimes \mathcal{R}}(x, r) := D_{\mathcal{X}}(x) \cdot D^q_{\mathcal{R}}(r).$$

The *conditional distribution* $D^q_{\mathcal{R}|\mathcal{X}}$ over $\mathcal{R}$ represents the probability to get $r$ as the output of $M(\cdot, q)$ from a sample set, given the fact that we got $x$ by sampling the same sample set uniformly at random. That is, $\forall x \in \mathcal{X}, r \in \mathcal{R}$,

$$D^q_{\mathcal{R}|\mathcal{X}}(r \mid x) := \sum_{s \in \mathcal{X}^n} D_{\mathcal{X}^n|\mathcal{X}}(s \mid x) \cdot D^q_{\mathcal{R}|\mathcal{X}^n}(r \mid s).$$

The *conditional distribution* $D^q_{\mathcal{X}|\mathcal{R}}$ over $\mathcal{X}$ represents the probability to get $x$ by sampling a sample set uniformly at random, given the fact that we got $r$ as the output of $M\left(\cdot, q\right)$ from that sample set. That is, $\forall x \in \mathcal{X}, r \in \mathcal{R}$,

$$D^q_{\mathcal{X}|\mathcal{R}}\left(x\,|\,r\right) \coloneqq \sum_{s \in \mathcal{X}^n} D^q_{\mathcal{X}^n|\mathcal{R}}\left(s\,|\,r\right) \cdot D_{\mathcal{X}|\mathcal{X}^n}\left(x\,|\,s\right).$$

Although all of these definitions depend on $D_{\mathcal{X}^n}$ and $M$, we typically omit these from the notation for simplicity, and usually omit the superscripts and subscripts entirely. We include them only when necessary for clarity. We also use $D$ to denote the probability of a set: for $\mathbf{r} \subseteq \mathcal{R}$, we define $D^q_{\mathcal{R}}\left(\mathbf{r}\right) \coloneqq \sum_{r \in \mathbf{r}} D^q_{\mathcal{R}}\left(r\right).$

Though the conditional distributions $D^q_{\mathcal{R}|\mathcal{X}}$ and $D^q_{\mathcal{X}|\mathcal{R}}$ were not defined as the ratio between the joint and marginal distribution, the analogue of Bayes' rule still holds for these distributions.

**Proposition A.3.** *Given any distribution $D_{\mathcal{X}^n}$, mechanism $M : \mathcal{X}^n \times \mathcal{Q} \to \mathcal{R}$, and query $q$,*

$$D_{(\mathcal{X},\mathcal{R})}\left(x, r\right) = D\left(x\right) \cdot D\left(r\,|\,x\right) = D\left(r\right) \cdot D\left(x\,|\,r\right).$$

*Proof.* We observe

$$\begin{aligned}
D_{(\mathcal{X},\mathcal{R})}\left(x, r\right) &= \sum_{s \in \mathcal{X}^n} D\left(s\right) \cdot D\left(x\,|\,s\right) \cdot D\left(r\,|\,s\right) \\
&= \sum_{s \in \mathcal{X}^n} D_{(\mathcal{X}^n, \mathcal{X})}\left(s, x\right) \cdot D\left(r\,|\,s\right) \\
&= D\left(x\right) \cdot \sum_{s \in \mathcal{X}^n} \frac{D_{(\mathcal{X}^n, \mathcal{X})}\left(s, x\right)}{D\left(x\right)} \cdot D\left(r\,|\,s\right) \\
&= D\left(x\right) \cdot \sum_{s \in \mathcal{X}^n} D\left(s\,|\,x\right) \cdot D\left(r\,|\,s\right) \\
&= D\left(x\right) \cdot D\left(r\,|\,x\right).
\end{aligned}$$

Similarly,

$$\begin{aligned}
D_{(\mathcal{X},\mathcal{R})}\left(x, r\right) &= \sum_{s \in \mathcal{X}^n} D\left(s\right) \cdot D\left(r\,|\,s\right) \cdot D\left(x\,|\,s\right) \\
&= \sum_{s \in \mathcal{X}^n} D_{(\mathcal{X}^n, \mathcal{R})}\left(s, r\right) \cdot D\left(x\,|\,s\right) \\
&= D\left(r\right) \cdot \sum_{s \in \mathcal{X}^n} \frac{D_{(\mathcal{X}^n, \mathcal{R})}\left(s, r\right)}{D\left(r\right)} \cdot D\left(x\,|\,s\right) \\
&= D\left(r\right) \cdot \sum_{s \in \mathcal{X}^n} D\left(s\,|\,r\right) \cdot D\left(x\,|\,s\right) \\
&= D\left(r\right) \cdot D\left(x\,|\,r\right).
\end{aligned}$$

$\square$

# B    Missing Details from Section 2

**Lemma B.1.** *Given $0 \leq \delta \leq \epsilon \leq 1$, a distribution $D_{\mathcal{X}^n}$, and a query $q$, if a mechanism $M$ is $(\epsilon, \delta)$-LSS with respect to $D_{\mathcal{X}^n}, q$, then $D\left(\mathbf{r}_{2\epsilon}\right) < \frac{\delta}{\epsilon}$.*

*Proof.* Assume by way of contradiction that $D\left(\mathbf{r}_{2\epsilon}\right) \geq \frac{\delta}{\epsilon}$; then

$$D\left(\mathbf{r}_{2\epsilon}\right) \cdot \left(\ell\left(\mathbf{r}_{2\epsilon}\right) - \epsilon\right) > \frac{\delta}{\epsilon} \cdot \left(2\epsilon - \epsilon\right) = \delta. \qquad \square$$

## B.1 Proof of Post-Processing Theorem

*Proof of Theorem 2.3.* We start by defining a function $w_{\mathcal{U}}^\epsilon : \mathcal{R} \to [0,1]$ such that $\forall r \in \mathcal{R} :$ $w_{\mathcal{U}}^\epsilon(r) = \sum_{u \in \mathbf{u}_\epsilon} D(u \,|\, r)$, where $\mathbf{u}_\epsilon$ is the set of $\epsilon$-unstable values in $\mathcal{U}$ as defined in Definition 2.1, and $D(u \,|\, r) := \Pr_{U \sim f(r)}[U = u \,|\, r]$. Using this function we get that,

$$\sum_{u \in \mathbf{u}_\epsilon} D(u) = \sum_{u \in \mathbf{u}_\epsilon} \sum_{r \in \mathcal{R}} D(r) \cdot D(u \,|\, r)$$

$$= \sum_{r \in \mathcal{R}} \overbrace{\sum_{u \in \mathbf{u}_\epsilon} D(u \,|\, r)}^{=w_{\mathcal{U}}^\epsilon(r)} \cdot D(r)$$

$$= \sum_{r \in \mathcal{R}} w_{\mathcal{U}}^\epsilon(r) \cdot D(r),$$

and

$$\sum_{u \in \mathbf{u}_\epsilon} D(u) \cdot \ell(u) = \sum_{u \in \mathbf{u}_\epsilon} D(u) \sum_{x \in \mathbf{x}_+(u)} (D(x \,|\, u) - D(x))$$

$$= \sum_{u \in \mathbf{u}_\epsilon} \sum_{x \in \mathbf{x}_+(u)} D(x)(D(u \,|\, x) - D(u))$$

$$= \sum_{u \in \mathbf{u}_\epsilon} \sum_{r \in \mathcal{R}} \sum_{x \in \mathbf{x}_+(u)} D(x)(D(r \,|\, x) - D(r)) D(u \,|\, r)$$

$$\overset{(1)}{\le} \sum_{u \in \mathbf{u}_\epsilon} \sum_{r \in \mathcal{R}} \sum_{x \in \mathbf{x}_+(r)} D(x)(D(r \,|\, x) - D(r)) D(u \,|\, r)$$

$$= \sum_{r \in \mathcal{R}} \overbrace{\sum_{u \in \mathbf{u}_\epsilon} D(u \,|\, r)}^{=w_{\mathcal{U}}^\epsilon(r)} \cdot D(r) \sum_{x \in \mathbf{x}_+(r)} (D(x \,|\, r) - D(x))$$

$$= \sum_{r \in \mathbf{r}} w_{\mathcal{U}}^\epsilon(r) \cdot D(r) \cdot \ell(r),$$

where (1) results from the definition of $\mathbf{x}_+(r)$.

Combining the two we get that

$$D(\mathbf{u}_\epsilon) \cdot (\ell(\mathbf{u}_\epsilon) - \epsilon) = \sum_{u \in \mathbf{u}_\epsilon} D(u)(\ell(u) - \epsilon)$$

$$\overset{(1)}{\le} \sum_{r \in \mathcal{R}} w_{\mathcal{U}}^\epsilon(r) \cdot D(r)(\ell(r) - \epsilon)$$

$$\overset{(2)}{\le} \sum_{r \in \mathbf{r}_\epsilon} \overbrace{w_{\mathcal{U}}^\epsilon(r)}^{\le 1} \cdot D(r)(\ell(r) - \epsilon)$$

$$\le \sum_{r \in \mathbf{r}_\epsilon} D(r)(\ell(r) - \epsilon)$$

$$\overset{(3)}{\le} \delta,$$

where (1) results from the two previous claims, (2) from the fact that we removed only negative terms and (3) from the LSS definition, which concludes the proof. $\qquad\square$

## B.2 Adaptivity and View-Induced Posterior Distributions

**Definition B.2** (View-induced posterior distributions). A sequence of mechanisms $\bar{M}$, an analyst $A$, and a view $v_k \in \mathcal{V}_k$ together induce a set of posterior distributions over $\mathcal{X}^n$, $\mathcal{X}$, and $\mathcal{R}$. For clarity we will denote these induced distributions by $P^{v_k}$ instead of $D$.

As mentioned before, all the distributions we consider stem from two basic distributions; the underlying distribution $D_{\mathcal{X}^n}$ and the conditional distribution $D^q_{\mathcal{R}|\mathcal{X}^n}$. The posteriors of these distributions change once we see $v_k$. $D_{\mathcal{X}^n}$ is replaced by $P^{v_k}_{\mathcal{X}^n} := D^A_{\mathcal{X}^n|\mathcal{V}_k}(\cdot \mid v_k)$ (actually, the rigorous notation should have been $P^{\bar{M},A,v_k}_{\mathcal{X}^n}$, but since $\bar{M}$ and $A$ will be fixed throughout this analysis, we omit them for simplicity). Similarly, $D^{q_{k+1}}_{\mathcal{R}|\mathcal{X}^n}(r \mid s)$ is replaced by

$$P^{v_k}_{\mathcal{R}|\mathcal{X}^n}(r \mid s) := D^{q_{k+1}}_{\mathcal{R}|\mathcal{X}^n}(r \mid s, v_k) = \Pr_{R \sim M_{k+1}(s, q_{k+1})}\left[R = r \mid s, \mathrm{Adp}_{\bar{M},k}(s, A) = v_k\right],$$

where $\mathrm{Adp}_{\bar{M},k}$ denotes the first $k$ iterations of the adaptive mechanism, which - as mentioned previously - determine the $k + 1$-th query.[10]

We next establish two important properties of the distributions over $\mathcal{V}_{k+1}$ induced by $\mathrm{Adp}_{\bar{M}}$ and their relation to the posterior distributions.

**Lemma B.3.** *Given a distribution $D_{\mathcal{X}^n}$, an analyst $A : \mathcal{V} \to \mathcal{Q}$, and a sequence of $k$ mechanisms $\bar{M}$, for any $v_{k+1} \in \mathcal{V}_{k+1}$ we denote $v_{k+1} = (v_k, r_{k+1})$. In this case, using notation from Definition B.2,*

$$D(v_{k+1}) = D(v_k) \cdot P^{v_k}(r_{k+1})$$

*and*

$$\ell^A_{D_{\mathcal{X}^n}}(v_{k+1}) \leq \ell^A_{D_{\mathcal{X}^n}}(v_k) + \ell^{q_{k+1}}_{P^{v_k}_{\mathcal{X}^n}}(r_{k+1}).$$

*Proof.* We begin by proving a set of relations between the prior distributions over $\mathcal{V}_{k+1}$ and the posterior distributions induced by the view $v_k$.

$$D_{(\mathcal{X}^n, \mathcal{V}_{k+1})}(s, v_{k+1}) = D(s) \cdot D(v_{k+1} \mid s)$$
$$\stackrel{(1)}{=} D(s) \cdot D(v_k \mid s) \cdot D^{q_{k+1}}(r_{k+1} \mid s, v_k)$$
$$= D(v_k) \cdot D(s \mid v_k) \cdot D^{q_{k+1}}(r_{k+1} \mid s, v_k)$$
$$= D(v_k) \cdot P^{v_k}(s) \cdot P^{v_k}(r_{k+1} \mid s)$$
$$= D(v_k) \cdot P^{v_k}_{(\mathcal{X}^n, \mathcal{R})}(s, r_{k+1}),$$

where (1) is a result of the fact that $q_{k+1}$ is a deterministic function of $v_k$. As mentioned in Definition B.2, the distribution of $r_{k+1}$ might depend on $v_k$ in the case of a stateful mechanism, but it is all encapsulated in the definition of $P$.

Using this identity and the definition of $P^{v_k}_{\mathcal{X}^n}$ we get that,

$$D(v_{k+1}) = \sum_{s \in \mathcal{X}^n} D_{(\mathcal{X}^n, \mathcal{V}_{k+1})}(s, v_{k+1}) = \sum_{s \in \mathcal{X}^n} D(v_k) \cdot P^{v_k}_{(\mathcal{X}^n, \mathcal{R})}(s, r_{k+1}) = D(v_k) \cdot P^{v_k}(r_{k+1}).$$

$$D(x \mid v_k) = \sum_{s \in \mathcal{X}^n} D(s \mid v_k) \cdot D(x \mid s) = \sum_{s \in \mathcal{X}^n} P^{v_k}(s) \cdot D(x \mid s) = P^{v_k}(x).$$

$$D(x \mid v_{k+1}) = \sum_{s \in \mathcal{X}^n} D(s \mid v_{k+1}) \cdot D(x \mid s) = \sum_{s \in \mathcal{X}^n} P^{v_k}(s \mid r_{k+1}) \cdot D(x \mid s) = P^{v_k}(x \mid r_{k+1}).$$

where we keep using the fact that $D\left(x\,|\,s\right)$ does not depend on the underlying distribution $D_{\mathcal{X}^n}$ at all. Using these identities we can analyze the stability loss, and we would do so by invoking an equivalent definition of the statistical distance (see Appendix F),

$$
\begin{aligned}
\ell^A_{D_{\mathcal{X}^n}}\left(v_{k+1}\right) &= \frac{1}{2}\sum_{x\in\mathcal{X}}\left|D\left(x\,|\,v_{k+1}\right) - D_{\mathcal{X}}\left(x\right)\right| \\
&\leq^{(1)} \frac{1}{2}\sum_{x\in\mathcal{X}}\left|D\left(x\,|\,v_k\right) - D_{\mathcal{X}}\left(x\right)\right| + \frac{1}{2}\sum_{x\in\mathcal{X}}\left|D\left(x\,|\,v_{k+1}\right) - D\left(x\,|\,v_k\right)\right| \\
&= \ell^A_{D_{\mathcal{X}^n}}\left(v_k\right) + \frac{1}{2}\sum_{x\in\mathcal{X}}\left|P^{v_k,q_{k+1}}_{\mathcal{X}|\mathcal{R}}\left(x\,|\,r_{k+1}\right) - P^{v_k}_{\mathcal{X}}\left(x\right)\right| \\
&= \ell^A_{D_{\mathcal{X}^n}}\left(v_k\right) + \ell^{q_{k+1}}_{P^{v_k}_{\mathcal{X}^n}}\left(r_{k+1}\right),
\end{aligned}
$$

where (1) is simply the triangle inequality. $\qquad\square$

### B.3 Proof of Linear Adaptive Composition Theorem

*Proof of Theorem 2.6.* This theorem is a direct result of combining Lemma B.3 with the triangle inequality over the posteriors created at any iteration, and the fact that the mechanisms are LSS over the new posterior distributions. Formally this is proven using induction on the number of adaptive iterations. The base case $k=0$ is the coin tossing step, which is independent of the set and therefore has zero loss. For the induction step we start by denoting the projections of $\mathbf{v}^{k+1}_{\epsilon_{[k+1]}}$ on $\mathcal{V}_k$ and $\mathcal{R}$ by

$$
\forall r_{k+1}\in\mathcal{R}, \mathbf{v}_k\left(r_{k+1}\right) := \left\{v_k\in\mathcal{V}_k\,|\,\left(v_k,r_{k+1}\right)\in\mathbf{v}^{k+1}_{\epsilon_{[k+1]}}\right\}
$$

$$
\forall v_k\in\mathcal{V}_k, \mathbf{r}\left(v_k\right) := \left\{r_{k+1}\in\mathcal{R}\,|\,\left(v_k,r_{k+1}\right)\in\mathbf{v}^{k+1}_{\epsilon_{[k+1]}}\right\},
$$

where $\epsilon_{[k]} := \sum_{i\in[k]}\epsilon_i$.

Using this notation and that in Definition B.2 we get that

$$
\begin{aligned}
D\left(\mathbf{v}^{k+1}_{\epsilon_{[k+1]}}\right) &\cdot \left(\ell^A_{D_{\mathcal{X}^n}}\left(\mathbf{v}^{k+1}_{\epsilon_{[k+1]}}\right) - \epsilon_{[k+1]}\right) \\
&= \sum_{v_{k+1}\in\mathbf{v}^{k+1}_{\epsilon_{[k+1]}}} D\left(v_{k+1}\right)\left(\ell^A_{D_{\mathcal{X}^n}}\left(v_{k+1}\right) - \epsilon_{[k+1]}\right) \\
&\stackrel{(1)}{\leq} \sum_{(v_k,r_{k+1})\in\mathbf{v}^{k+1}_{\epsilon_{[k+1]}}} D\left(v_k\right)\cdot P^{v_k}\left(r_{k+1}\right)\left(\ell^A_{D_{\mathcal{X}^n}}\left(v_k\right) + \ell^{q_{k+1}}_{P^{v_k}_{\mathcal{X}^n}}\left(r_{k+1}\right) - \epsilon_{[k+1]}\right)
\end{aligned}
$$

where (1) is a direct result of Lemma B.3. Analyzing the two parts separately we get

$$
\begin{aligned}
\sum_{v_k\in\mathcal{V}_k}\sum_{r_{k+1}\in\mathbf{r}(v_k)} D\left(v_k\right)\cdot P^{v_k}\left(r_{k+1}\right)\left(\ell^A_{D_{\mathcal{X}^n}}\left(v_k\right) - \epsilon_{[k]}\right) &\stackrel{(1)}{\leq} \sum_{v_k\in\mathbf{v}^k_{\epsilon_{[k]}}} D\left(v_k\right)\left(\ell^A_{D_{\mathcal{X}^n}}\left(v_k\right) - \epsilon_{[k]}\right) \\
&= D\left(\mathbf{v}^k_{\epsilon_{[k]}}\right)\left(\ell^A_{D_{\mathcal{X}^n}}\left(\mathbf{v}^k_{\epsilon_{[k]}}\right) - \epsilon_{[k]}\right) \\
&\stackrel{(2)}{\leq} \sum_{i\in[k]}\delta_i
\end{aligned}
$$

and similarly,

$$\sum_{r_{k+1}\in\mathcal{R}}\sum_{v_k\in\mathbf{v}_k(r_{k+1})} D\left(v_k\right)\cdot P^{v_k}\left(r_{k+1}\right)\left(\ell_{P_{\mathcal{X}^n}^{v_k}}^{q_{k+1}}\left(r_{k+1}\right)-\epsilon_{k+1}\right)$$

$$\overset{(1)}{\leq}\sum_{r_{k+1}\in\mathbf{r}_{\epsilon_{k+1}}^{q_{k+1}}} P^{v_k}\left(r_{k+1}\right)\left(\ell_{P_{\mathcal{X}^n}^{v_k}}^{q_{k+1}}\left(r_{k+1}\right)-\epsilon_{k+1}\right)$$

$$=P^{v_k}\left(\mathbf{r}_{\epsilon_{k+1}}^{q_{k+1}}\right)\left(\ell_{P_{\mathcal{X}^n}^{v_k}}^{q_{k+1}}\left(\mathbf{r}_{\epsilon_{k+1}}^{q_{k+1}}\right)-\epsilon_{k+1}\right)$$

$$\overset{(2)}{\leq}\delta_{k+1},$$

where in both cases (1) is a result of the fact that in both sums we add positive summands and remove negative ones, and (2) results from the inductive assumption.

Combining the two we get that $D\left(\mathbf{v}_{\epsilon_{[k+1]}}^{k+1}\right)\cdot\left(\ell_{D_{\mathcal{X}^n}}^{A}\left(\mathbf{v}_{\epsilon_{[k+1]}}^{k+1}\right)-\epsilon_{[k+1]}\right)\leq\sum_{i\in[k+1]}\delta_i.$  $\qquad\square$

### B.4 Proof of Sub-Linear Adaptive Composition Theorem

**Lemma B.4** (Azuma inequality extended to high probability bound). *Given $k\in\mathbb{N}$, $0\leq\epsilon_1,\ldots,\epsilon_k$, $0\leq\delta_1,\ldots,\delta_k\leq 1$, if $Y_0,\ldots,Y_k$ is a martingale with respect to another sequence $Z_0,\ldots,Z_k$ such that for any $i\in[k]$, $Pr\left[|Y_i-Y_{i-1}|>\epsilon_i\right]\leq\delta_i$, then for any $\lambda>0$,*

$$Pr\left[|Y_k-Y_0|>\lambda\right]\leq\exp\left(-\frac{\lambda^2}{2\sum_{i=1}^{k}\epsilon_i^2}\right)+\sum_{i=1}^{k}\delta_i.$$

The proof parallels that of a similar lemma by [TV$^+$15] (their Proposition 34).

*Proof.* For any given realization of the random variable $y=(y_0,\ldots,y_k)$, we denote by $I\left(y\right)$ the first index $i$ for which $|y_i-y_{i-1}|>\epsilon_i$. If no such index exists, $I\left(y\right)=k+1$. We then define $y':=f\left(y\right)$ where $\forall i<I\left(\bar{y}\right):y_i'=y_i$ and $\forall i\geq I\left(\bar{y}\right):y_i'=y_{I(\bar{y})-1}$. Notice that the random variable $Y'$ is also a martingale with respect to $Z_0$, $\Pr\left[\left|Y_i'-Y_{i-1}'\right|>\epsilon_i\right]=0$, and

$$\Pr\left[Y'\neq Y\right]\leq\sum_{i=1}^{k}\Pr\left[Y_i'\neq Y_i\right]\leq\sum_{i=1}^{k}\delta_i.$$

Using these facts we get

$$\Pr\left[|Y_k-Y_0|>\lambda\right]=\overbrace{\Pr\left[Y'=Y\right]}^{\leq 1}\cdot\Pr\left[|Y_k'-Y_0'|>\lambda\right]+\Pr\left[Y'\neq Y\right]\cdot\overbrace{\Pr\left[|Y_k-Y_0|>\lambda\right]}^{\leq 1}$$

$$\overset{(1)}{\leq}\exp\left(-\frac{\lambda^2}{2\sum_{i=1}^{k}\epsilon_i^2}\right)+\sum_{i=1}^{k}\delta_i.$$

where (1) results from the previous inequality and Azuma's inequality for $Y'$.  $\qquad\square$

*Proof of Theorem 2.7.* The proof is based on the fact that the sum of the stability losses is a martingale with respect to $v_k$, and invoking Lemma B.4.

Formally, for any given $k>0$, we can define $\Omega_0:=\mathcal{X}^n$ and $\forall i\in[k],\Omega_i:=\mathcal{R}.$[11] We define a probability distribution over $\Omega_0$ as $D_{\mathcal{X}^n}$, and for any $i>0$, define a probability distribution over $\Omega_i$

given $\Omega_1, \ldots, \Omega_{i-1}$ as $P^{v_{i-1}}$ (see Definition B.2). We then define a sequence of functions, $y_0 = 0$ and $\forall i > 0$,

$$y_i\left(s, r_1, \ldots, r_i\right) = \sum_{j=1}^{i} \left( \ell_{P^{v_{j-1}}}\left(r_j\right) - \mathop{\mathbb{E}}_{R \sim P_{\mathcal{R}}^{v_{j-1}}} \left[\ell_{P^{v_{j-1}}}\left(R\right)\right] \right).$$

Intuitively $y_i$ is the sum of the first $i$ losses, with a correction term which zeroes the expectation.

Notice that these random variables are a martingale with respect to the random process $S, R_1, \ldots, R_k$ since

$$
\mathop{\mathbb{E}}_{R_{i+1}}\left[Y_{i+1} \mid S, R_1, \ldots, R_i\right] = \mathop{\mathbb{E}}_{R_{i+1}}\left[ \overbrace{\sum_{j=1}^{i+1} \left( \ell_{P^{V_{j-1}}}\left(R_j\right) - \mathop{\mathbb{E}}_{R \sim P_{\mathcal{R}}^{V_{j-1}}}\left[\ell_{P^{V_{j-1}}}\left(R\right)\right]\right)}^{} \mid S, R_1, \ldots, R_i \right]
$$

$$
= \overbrace{\sum_{j=1}^{i} \left( \ell_{P^{V_{j-1}}}\left(R_j\right) - \mathop{\mathbb{E}}_{R \sim P_{\mathcal{R}}^{V_{j-1}}}\left[\ell_{P^{V_{j-1}}}\left(R\right)\right]\right)}^{=Y_i(S,R_1,\ldots,R_i)}
$$

$$
+ \overbrace{\mathop{\mathbb{E}}_{R_{i+1}}\left[\ell_{P^{V_i}}\left(R_{i+1}\right) - \mathop{\mathbb{E}}_{R \sim P_{\mathcal{R}}^{V_i}}\left[\ell_{P^{V_i}}\left(R\right)\right] \mid S, R_1, \ldots, R_i\right]}^{=0}
$$

$$
= Y_i\left(S, R_1, \ldots, R_i\right)
$$

where the expectation is taken over the random process, which has randomness that results from the choice of $s \in \mathcal{X}^n$ and the internal probability of $M$.

From the LSS definition (Definition 2.2) and Lemma B.1, for any $i \in [k]$ we get that $\mathop{\Pr}_{R \sim P_{\mathcal{R}}^{v_{i-1}}}\left[\ell_{P^{v_i}}\left(R_i\right) > 2\epsilon_i\right] \leq \frac{\delta_i}{\epsilon_i}$, so with probability greater than $\frac{\delta_{i+1}}{\epsilon_{i+1}}$,

$$
\left|Y_{i+1} - Y_i\right| = \left| \ell_{P^{V_i}}\left(R_{i+1}\right) - \mathop{\mathbb{E}}_{R \sim P^{V_i}}\left[\ell_{P^{V_i}}\left(R\right)\right] \right| \leq \ell_{P^{V_i}}\left(R_{i+1}\right) \leq 2\epsilon_{i+1}.
$$

Using this fact we can invoke Lemma B.4 and get that for any $0 \leq \delta' \leq 1$,

$$\mathop{\Pr}_{V \sim D_{\mathcal{V}_k}}\left[\ell_{D_{\mathcal{X}^n}}\left(V\right) > \epsilon'\right]$$

$$
\stackrel{(1)}{\leq} \mathop{\Pr}_{V \sim D_{\mathcal{V}_k}}\left[ \sum_{i=1}^{k} \ell_{P^{V_{i-1}}}\left(R_i\right) > \sqrt{8\ln\left(\frac{1}{\delta'}\right)\sum_{i=1}^{k}\epsilon_i^2} + \sum_{i=1}^{k}\alpha_i \right]
$$

$$
\stackrel{(2)}{\leq} \mathop{\Pr}_{V \sim D_{\mathcal{V}_k}}\left[ \sum_{j=1}^{k} \left( \ell_{P^{V_{j-1}}}\left(R_j\right) - \mathop{\mathbb{E}}_{R \sim P_{\mathcal{R}}^{V_{j-1}}}\left[\ell_{P^{V_{j-1}}}\left(R\right)\right]\right) > \sqrt{8\ln\left(\frac{1}{\delta'}\right)\sum_{i=1}^{k}\epsilon_i^2} \right]
$$

$$
\stackrel{(3)}{=} \mathop{\Pr}_{V \sim D_{\mathcal{V}_k}}\left[ Y_k - \overbrace{Y_0}^{=0} > \sqrt{8\ln\left(\frac{1}{\delta'}\right)\sum_{i=1}^{k}\epsilon_i^2} \right]
$$

$$
\stackrel{(4)}{\leq} \delta' + \sum_{i=1}^{k}\frac{\delta_i}{\epsilon_i}
$$

where (1) results from Lemma B.3, (2) from the bound on the expectation of the stability loss, (3) from the definition of $Y_i$, and (4) from Lemma B.4. $\square$

## C  Missing Details from Section 3

### C.1  Generalization of Expectation

As a step toward showing that LS Stability implies a high probability generalization, we first show a generalization of expectation result. We do so, as a tool, specifically for a mechanism that returns

a query as its output. Intuitively, this allows us to wrap an entire adaptive process into a single mechanism. Analyzing the potential of the mechanism to generate an overfitting query is a natural way to learn about the generalization capabilities of the mechanism.

**Theorem C.1** (Generalization of expectation). *Given $0 \le \epsilon, \delta \le 1$, a distribution $D_{\mathcal{X}^n}$, a query $q$, and a mechanism $M : \mathcal{X}^n \times \mathcal{Q} \to \mathcal{Q}_\Delta$, if $D\left(\mathbf{q}_\epsilon\right) < \delta$, then*

$$\left| \mathop{\mathbb{E}}_{S \sim D_{\mathcal{X}^n}, Q' \sim M(S,q)} \left[ Q'\left(D_{\mathcal{X}^n}\right) - Q'\left(S\right) \right] \right| < 2\Delta\left(\epsilon + \delta\right).$$

*Proof.* First notice that,

$$q\left(s\right) = \frac{1}{n}\sum_{i=1}^{n} q\left(s_i\right) = \sum_{x \in \mathcal{X}} D\left(x \mid s\right) \cdot q\left(x\right)$$

where $s_1, \dots s_n$ denotes the elements of the sample set $s$. Using this identity we separately analyze the expected value of the returned query with respect to the distribution, and with respect to the sample set.

$$\mathop{\mathbb{E}}_{S \sim D_{\mathcal{X}^n}, Q' \sim M(S,q)} \left[Q'\left(D_{\mathcal{X}^n}\right)\right] = \sum_{s \in \mathcal{X}^n} D\left(s\right) \cdot \sum_{q' \in \mathcal{Q}_\Delta} D\left(q' \mid s\right) \cdot q'\left(D_{\mathcal{X}^n}\right)$$

$$= \sum_{q' \in \mathcal{Q}_\Delta} \sum_{s \in \mathcal{X}^n} D\left(s\right) \cdot \overbrace{D\left(q' \mid s\right)}^{=D\left(q'\right)} \cdot \overbrace{\sum_{s' \in \mathcal{X}^n} D_{\mathcal{X}^n}\left(s'\right) \cdot q\left(s'\right)}^{=q'\left(D_{\mathcal{X}^n}\right)}$$

$$= \sum_{q' \in \mathcal{Q}_\Delta} D\left(q'\right) \sum_{x \in \mathcal{X}} \overbrace{\sum_{s' \in \mathcal{X}^n} D\left(s'\right) \cdot D\left(x \mid s'\right)}^{D(x)} \cdot q'\left(x\right)$$

$$= \sum_{q' \in \mathcal{Q}_\Delta} D\left(q'\right) \sum_{x \in \mathcal{X}} D\left(x\right) \cdot q'\left(x\right)$$

$$\mathop{\mathbb{E}}_{S \sim D_{\mathcal{X}^n}, Q' \sim M(S,q)} \left[Q'\left(S\right)\right] = \sum_{s \in \mathcal{X}^n} D\left(s\right) \cdot \sum_{q' \in \mathcal{Q}_\Delta} D\left(q' \mid s\right) \cdot q'\left(s\right)$$

$$= \sum_{q' \in \mathcal{Q}_\Delta} \sum_{x \in \mathcal{X}} \overbrace{\sum_{s \in \mathcal{X}^n} D\left(s\right) \cdot D\left(q' \mid s\right) \cdot D\left(x \mid s\right)}^{=D^q_{\left(\mathcal{X}, \mathcal{Q}_\Delta\right)}\left(x, q'\right)} \cdot q'\left(x\right)$$

$$\overset{(1)}{=} \sum_{q' \in \mathcal{Q}_\Delta} D\left(q'\right) \sum_{x \in \mathcal{X}} D\left(x \mid q'\right) \cdot q'\left(x\right),$$

where (1) is a result of Lemma A.3.

Now we can calculate the difference:

$$\left| \mathop{\mathbb{E}}_{S \sim D_{\mathcal{X}^n}, Q' \sim M(S,q)} \left[Q'\left(D_{\mathcal{X}^n}\right) - Q'\left(S\right)\right] \right| = \left| \sum_{q' \in \mathcal{Q}_\Delta} D\left(q'\right) \sum_{x \in \mathcal{X}} \left(D\left(x\right) - D\left(x \mid q'\right)\right) \cdot q'\left(x\right) \right|$$

$$\overset{(1)}{\le} \sum_{q' \in \mathcal{Q}_\Delta} D\left(q\right) \overbrace{\sum_{x \in \mathcal{X}} \left|D\left(x\right) - D\left(x \mid q'\right)\right|}^{=2\ell\left(q'\right)} \cdot \Delta$$

$$= 2\Delta \cdot \left( \overbrace{\sum_{q' \notin \mathbf{q}_\epsilon} D\left(q'\right)}^{\le 1} \cdot \overbrace{\ell\left(q'\right)}^{\le \epsilon} + \overbrace{\sum_{q' \in \mathbf{q}_\epsilon} D\left(q'\right)}^{<\delta} \cdot \overbrace{\ell\left(q'\right)}^{\le 1} \right)$$

$$\overset{(2)}{<} 2\Delta\left(\epsilon + \delta\right),$$

where (1) results from the definition of $\mathcal{Q}_\Delta$ and the triangle inequality, and (2) from the condition that $D(\mathbf{q}_\epsilon) < \delta$. $\square$

**Corollary C.2.** *Given* $0 \leq \epsilon, \delta \leq 1$, *a distribution* $D_{\mathcal{X}^n}$, *and a query* $q$, *if a mechanism* $M : \mathcal{X}^n \times \mathcal{Q} \to \mathcal{Q}_\Delta$ *is* $(\epsilon, \delta)$-*LSS with respect to* $D_{\mathcal{X}^n}, q$, *then*

$$\left| \mathop{\mathbb{E}}_{S \sim D_{\mathcal{X}^n}, Q' \sim M(S,q)} [Q'(D_{\mathcal{X}^n}) - Q'(S)] \right| < 2\Delta \left( 2\epsilon + \frac{\delta}{\epsilon} \right).$$

*Proof.* This is a direct result of combining Theorem C.1 with Lemma B.1. $\square$

## C.2 Monitoring Argument

**Definition C.3** (The Monitor Mechanism)**.** The *Monitor Mechanism* is a function $\mathrm{Mon}_{\bar{M}} : (\mathcal{X}^n)^t \times \mathcal{A} \to \mathcal{Q} \times \mathbb{R} \times [t]$ which is parametrized by a sequence of $k$ mechanisms $\bar{M}$ where $\forall i \in [k]$, $M_i : \mathcal{X}^n \times \mathcal{Q} \to \mathbb{R}$. Given a series of sample sets $\bar{s} \in (\mathcal{X}^n)^t$ and analyst $A$ as input, it runs the adaptive mechanism between $\bar{M}$ and $A$ for $t$ independent times (which in particular means neither of them share state across those iterations) and outputs a query $q \in \mathcal{Q}$, response $r \in \mathbb{R}$ and index $i \in t$, based on the following process:

---

Monitor Mechanism $\mathrm{Mon}_{\bar{M}}$

---

**Input:** $\bar{s} \in (\mathcal{X}^n)^t, A \in \mathcal{A}$
**Output:** $q \in \mathcal{Q}, r \in \mathbb{R}, i \in t$
**for** $i = 1, ..., t$ :
   $v^i \leftarrow \mathrm{Adp}_{\bar{M}}(s_i, A)$
   $(\tilde{q}^i, \tilde{r}^i) \leftarrow \mathop{\arg\max}\limits_{(q,r) \in v^i} |q(D_{\mathcal{X}^n}) - r|^a$
  **if** $\tilde{q}^i(D_{\mathcal{X}^n}) \geq \tilde{r}^i$:[b]
   $q^i \leftarrow \tilde{q}^i$
   $r^i \leftarrow \tilde{r}^i$
  **else**:
   $q^i \leftarrow -\tilde{q}^i$
   $r^i \leftarrow -\tilde{r}^i$
$i^* \leftarrow \mathop{\arg\max}\limits_{i \in [t]} \left( q^i(D_{\mathcal{X}^n}) - r^i \right)$
**return** $\left( q^{i^*}, r^{i^*}, i^* \right)$

---

  [a]We slightly abuse notation since $q$ is not part of $v^i$, but since it can be recovered from it, this term is well defined.

  [b]The addition of this condition ensures that $q(D_{\mathcal{X}^n}) \geq r$ for the output of the mechanism, a fact that will be used later in the proof of Claim C.6.

---

Notice that the monitor mechanism makes use of the ability to evaluate queries according to the true underlying distribution.[12]

We begin by proving that the monitor mechanism has generalization of expectation. In this claim and the following ones, the probabilities and expectations are taken over the randomness of the choice of $\bar{s} \in (\mathcal{X}^n)^t$ (which is assumed to be drawn iid from $D_{\mathcal{X}^n}$) and the internal probability of $\mathrm{Adp}_{\bar{M}}$.

**Claim C.4.** *Given* $0 \leq \epsilon, \delta \leq 1$, $t \in \mathbb{N}$, *a distribution* $D_{\mathcal{X}^n}$, *and an analyst* $A : \mathbb{V} \to \mathcal{Q}_\Delta$, *if a sequence of* $k$ *mechanisms* $\bar{M}$ *where* $\forall i \in [k]$, $M_i : \mathcal{X}^n \times \mathcal{Q}_\Delta \to \mathbb{R}$ *is* $(\epsilon, \delta)$-$k$-*LSS with respect to* $D_{\mathcal{X}^n}, A$, *then*

$$\left| \mathop{\mathbb{E}}_{\bar{S} \sim D_{\mathcal{X}^n}^t, (Q,R,I) \sim Mon_{\bar{M}}(\bar{S},A)} [q(D_{\mathcal{X}^n}) - Q(S_I)] \right| < 2\Delta \left( 2\epsilon + \frac{t\delta}{\epsilon} \right).$$

*Proof.* Since $q^i$ is a post-processing of $v^i$ and $\mathrm{Adp}_{\bar{M}}$ is $(\epsilon, \delta)$-LSS with respect to $A$, Theorem 2.3 implies that the post-processing producing $q^i$ is $(\epsilon, \delta)$-LSS with respect to $A$ as well. Using Lemma

B.1 we get that $D\left(\mathbf{q}_{2\epsilon}\right) < \frac{\delta}{\epsilon}$ for each of the $t$ rounds. Using the union bound and the fact that the $t$ rounds are independent we get that $\Pr_{\bar{S}\sim D_{\mathcal{X}^n}^t,(Q,R,I)\sim\mathrm{Mon}_{\bar{M}}\left(\bar{S},A\right)}[q \in \mathbf{q}_{2\epsilon}] < \frac{t\delta}{\epsilon}$. This allows us to invoke Theorem C.1, with $\frac{t\delta}{\epsilon}$ replacing $\delta$.[13] $\qquad\square$

We next show that $k$-sample accuracy of the mechanism run inside the monitor mechanism has implications for the sample accuracy of the result of the monitor mechanism.

**Claim C.5.** *Given* $0 \leq \epsilon, \delta \leq 1$, $t \in \mathbb{N}$, *a distribution* $D_{\mathcal{X}^n}$, *and an analyst* $A : \mathbb{V} \to \mathcal{Q}_\Delta$, *if a sequence of* $k$ *mechanisms* $\bar{M}$ *where* $\forall i \in [k]$, $M_i : \mathcal{X}^n \times \mathcal{Q}_\Delta \to \mathbb{R}$ *is* $(\epsilon, \delta)$-$k$-*Sample Accurate with respect to* $D_{\mathcal{X}^n}$, $A$, *then*

$$\mathbb{E}_{\bar{S}\sim D_{\mathcal{X}^n}^t,(Q,R,I)\sim Mon_{\bar{M}}\left(\bar{S},A\right)}\left[Q\left(S_I\right) - R\right] \leq \epsilon + 2t\delta\Delta.$$

*Proof.* This is a direct result of combining the sample accuracy definition and the union bound. If the probability that the sample accuracy of $M$ will be greater than $\epsilon$ is bounded by $\delta$, then the probability that it will fail to hold once in $t$ independent iterations is less then $t\delta$, and since the values of the query are bounded on the interval $[-\Delta, \Delta]$ the maximal error in these cases is $2\Delta$. $\qquad\square$

If the mechanism run by the monitor mechanism is not $k$-Distribution Accurate, this has implications for the distribution accuracy of the result of the monitor mechanism, as well.

**Claim C.6.** *Given* $0 \leq \epsilon, \delta \leq 1$, $t \in \mathbb{N}$, *a distribution* $D_{\mathcal{X}^n}$, *and an analyst* $A : \mathbb{V} \to \mathcal{Q}_\Delta$, *if a a sequence of* $k$ *mechanisms* $\bar{M}$ *where* $\forall i \in [k]$, $M_i : \mathcal{X}^n \times \mathcal{Q}_\Delta \to \mathbb{R}$ *is* **not** $(\epsilon, \delta)$-$k$-*Distribution Accurate with respect to* $D_{\mathcal{X}^n}$, $A$, *then*

$$\mathbb{E}_{\bar{S}\sim D_{\mathcal{X}^n}^t,(Q,R,I)\sim Mon_{\bar{M}}\left(\bar{S},A\right)}\left[Q\left(D_{\mathcal{X}^n}\right) - R\right] > \epsilon\left(1 - (1-\delta)^t\right).$$

*Proof.* First recall that from the definition of the monitor mechanism, $\forall i \in [t]$, $q_i\left(D_{\mathcal{X}^n}\right) - r_i \geq 0$. Therefore if $M$ is not $(\epsilon, \delta)$-Distribution Accurate, then $\forall i \in [t]$

$$\Pr_{S\sim D_{\mathcal{X}^n}^t, V\sim M_i(S,A),(Q,R)=\arg\max_{(q,r)\in V}|q(D_{\mathcal{X}^n})-r|}\left[Q\left(D_{\mathcal{X}^n}\right) - R > \epsilon\right] > \delta.$$

Since the $t$ rounds of the monitor mechanism are independent and $i^*$ is the index of the round with the maximal error,

$$\Pr_{\bar{S}\sim D_{\mathcal{X}^n}^t,(Q,R,I)\sim\mathrm{Mon}_{\bar{M}}\left(\bar{S},A\right)}\left[Q\left(D_{\mathcal{X}^n}\right) - R > \epsilon\right] > 1 - (1-\delta)^t.$$

So the expectation of this quantity must be greater then $\epsilon\left(1 - (1-\delta)^t\right)$, concluding the proof. $\qquad\square$

Finally, we use the monitor mechanism as a tool to show that LSS implies generalization with high probability.

### C.3 Proof of Generalization in Probability Theorem

*Proof of Theorem 3.3.* We will prove a slightly more general claim. For every $0 < a, b, c, d$ such that $4a + 2b + c + 2d < 1 - e^{-1}$, say $M$ is both $\left(a\frac{\epsilon}{\Delta}, ab\frac{\epsilon^2\delta}{\Delta^2}\right)$-$k$-LSS and $\left(c\epsilon, d\frac{\epsilon\delta}{\Delta}\right)$-$k$-Sample Accurate and assume $M$ is not $(\epsilon, \delta)$-$k$-Distribution Accurate.

Setting $t = \lfloor \frac{1}{\delta} \rfloor$, we see

$$\left| \underset{\bar{S} \sim D_{\mathcal{X}^n}^t, (Q,R,I) \sim \mathrm{Mon}_{\bar{M}}(\bar{S},A)}{\mathbb{E}} [Q(D_{\mathcal{X}^n}) - Q(S_I)] \right| \overset{(1)}{<} 2\Delta \left( 2\frac{a\epsilon}{\Delta} + \frac{t\Delta}{a\epsilon} \cdot \frac{ab\epsilon^2\delta}{\Delta^2} \right)$$

$$\overset{(2)}{\leq} (4a + 2b)\epsilon,$$

where (1) is a direct result of Claim C.4 and (2) uses the definition of $t$.

But on the other hand,

$$\left| \underset{\bar{S} \sim D_{\mathcal{X}^n}^t, (Q,R,I) \sim \mathrm{Mon}_{\bar{M}}(\bar{S},A)}{\mathbb{E}} [Q(D_{\mathcal{X}^n}) - Q(S_I)] \right| \overset{(1)}{\geq} \left| \underset{\bar{S} \sim D_{\mathcal{X}^n}^t, (Q,R,I) \sim \mathrm{Mon}_{\bar{M}}(\bar{S},A)}{\mathbb{E}} [Q(D_{\mathcal{X}^n}) - R] \right|$$

$$- \left| \underset{\bar{S} \sim D_{\mathcal{X}^n}^t, (Q,R,I) \sim \mathrm{Mon}_{\bar{M}}(\bar{S},A)}{\mathbb{E}} [Q(S_I) - R] \right|$$

$$\overset{(2)}{>} \epsilon\left(1 - (1-\delta)^t\right) - \left(c\epsilon + 2t \cdot \frac{d\epsilon\delta}{\Delta}\Delta\right)$$

$$\overset{(3)}{>} \epsilon\left(1 - e^{-\delta\lfloor \frac{1}{\delta} \rfloor}\right) - (c + 2d)\epsilon$$

$$\overset{(4)}{\geq} \epsilon\left(1 - e^{-1}\right) - (c + 2d)\epsilon$$

$$\overset{(5)}{>} (4a + 2b)\epsilon,$$

where (1) is the triangle inequality, (2) uses Claims C.5 and C.6, (3) the definition of $t$, (4) the inequality $1 - \delta \leq e^{-\delta}$, and (5) the definition of $a, b, c, d$. Since combining all of the above leads to a contradiction, we know that $\bar{M}$ must be $(\epsilon, \delta)$-Distribution Accurate, which concludes the proof. The theorem was stated choosing $a = c = \frac{1}{8}, b = d = \frac{1}{600}$. $\qquad\square$

### C.4 Proof of the Necessity of LSS to Generalization Theorem

**Definition C.7** (The Second Monitor Mechanism). The *Second Monitor Mechanism* is a function $\mathrm{Mon2}_{\bar{M}} : (\mathcal{X}^n)^t \times \mathcal{A} \to \mathcal{Q} \times \mathbb{R} \times [t]$ which is parametrized by a sequence of $k$ mechanisms $\bar{M}$ where $\forall i \in [k], M_i : \mathcal{X}^n \times \mathcal{Q} \to \mathbb{R}$. Given a series of sample sets $\bar{s} \in (\mathcal{X}^n)^t$ and analyst $A$ as input, it runs the adaptive mechanism between $\bar{M}$ and $A$ for $t$ independent times and outputs a query $q \in \mathcal{Q}$, response $r \in \mathbb{R}$ and index $i \in t$, based on the following process:

---
Second Monitor Mechanism $\mathrm{Mon2}_{\bar{M}}$

**Input:** $\bar{s} \in (\mathcal{X}^n)^t, A \in \mathcal{A}$
**Output:** $q \in \mathcal{Q}, r \in \mathbb{R}, i \in t$
**for** $i = 1, ..., t$ :
$\quad v^i \leftarrow \mathrm{Adp}_{\bar{M}}(s_i, A)$
$\quad q^i \leftarrow \tilde{q}_{v^i}$
$\quad r^i \leftarrow M(s, q^i)$
$i^* \leftarrow \underset{i \in [t]}{\arg\max}\left(\ell^A_{D_{\mathcal{X}^n}}(v^i)\right)$
**return** $\left(q^{i^*}, r^{i^*}, i^*\right)$

---

*Proof of Theorem 3.6.* Again we will prove a slightly more general claim. For every $0 < a, b, c, d$ such that $a + 2b + c + 2d < 2\left(1 - e^{-1}\right)$, say $M$ is both $\left(a\epsilon, b\frac{\epsilon\delta}{\Delta}\right)$ $(k+1)$-Sample Accurate and $\left(c\epsilon, d\frac{\epsilon\delta}{\Delta}\right)$ $(k+1)$-Distribution Accurate and assume $M$ is not $\left(\frac{\epsilon}{\Delta}, \delta\right)$-$k$-LSS.

First notice that if $\bar{M}$ is not $\left(\frac{\epsilon}{\Delta}, \delta\right)$-$k$-LSS with respect to $D_{\mathcal{X}^n}, A$, then in particular $D\left(\mathbf{v}^k_{\left(\frac{\epsilon}{\Delta}\right)}\right) \geq \delta$.

Since the $t$ rounds of the second monitor mechanism are independent and $i^*$ is the index of the round with the maximal stability loss of the calculated query, we get that

$$\underset{\bar{S} \sim D_{\mathcal{X}^n}^t, (Q,R,I) \sim \mathrm{Mon2}_{\bar{M}}(\bar{S},A)}{\Pr}\left[v^I \in \mathbf{v}^k_{\left(\frac{\epsilon}{\Delta}\right)}\right] > 1 - (1-\delta)^t.$$

Combining this fact with Lemma 3.5, and setting $t = \lfloor \frac{1}{\delta} \rfloor$ we get on one hand,

$$
\left| \mathop{\mathbb{E}}_{\bar{S} \sim D_{\mathcal{X}^n}^t, (Q,R,I) \sim \mathrm{Mon2}_{\bar{M}}(\bar{S}, A)} [Q(D_{\mathcal{X}^n}) - Q(S_I)] \right| \overset{(1)}{\geq} 2 \frac{\epsilon}{\Delta} \Delta \left( 1 - (1-\delta)^t \right)
$$

$$
\overset{(2)}{>} 2\epsilon \left( 1 - e^{-\delta \lfloor \frac{1}{\delta} \rfloor} \right)
$$

$$
\overset{(3)}{>} 2\epsilon \left( 1 - e^{-1} \right),
$$

where (1) is a direct result of invoking Lemma 3.5 with $1 - (1-\delta)^t$ for $\delta$, (2) uses the definition of $t$ and (3) uses the inequality $1 - \delta \leq e^{-\delta}$.

But on the other hand,

$$
\left| \mathop{\mathbb{E}}_{\bar{S} \sim D_{\mathcal{X}^n}^t, (Q,R,I) \sim \mathrm{Mon2}_{\bar{M}}(\bar{S}, A)} [Q(D_{\mathcal{X}^n}) - Q(S_I)] \right| \overset{(1)}{\leq} \left| \mathop{\mathbb{E}}_{\bar{S} \sim D_{\mathcal{X}^n}^t, (Q,R,I) \sim \mathrm{Mon2}_{\bar{M}}(\bar{S}, A)} [Q(D_{\mathcal{X}^n}) - R] \right|
$$

$$
+ \left| \mathop{\mathbb{E}}_{\bar{S} \sim D_{\mathcal{X}^n}^t, (Q,R,I) \sim \mathrm{Mon2}_{\bar{M}}(\bar{S}, A)} [Q(S_I) - R] \right|
$$

$$
\overset{(2)}{<} \left( a\epsilon + 2t \cdot \frac{b\epsilon\delta}{\Delta} \Delta \right) + \left( c\epsilon + 2t \cdot \frac{d\epsilon\delta}{\Delta} \Delta \right)
$$

$$
\overset{(3)}{\leq} (a + 2b + c + 2d) \epsilon
$$

$$
\overset{(4)}{<} 2\epsilon \left( 1 - e^{-1} \right),
$$

where (1) is the triangle inequality, (2) uses Claim C.5 which was mentioned with relation to the original monitor mechanism (this time for the distribution error as well), (3) uses the definition of $t$, and (4) the definition of $a, b, c, d$.

Since combining all of the above leads to a contradiction, we know that $\bar{M}$ cannot be $\left( \frac{\epsilon}{\Delta}, \delta \right)$-$k$-LSS, which concludes the proof. The theorem was stated choosing $a = b = c = d = \frac{1}{5}$.  □

# D   Missing Details from Section 4

## D.1   Definitions

In the following definitions, $\mathcal{X}, D_{\mathcal{X}}, \mathcal{Q}, \mathcal{R}, M, \epsilon, \delta$ and $n$ are used in a similar manner as for the definitions leading to LSS.

**Definition D.1** (Differential Privacy [DMNS06]). Given $0 \leq \epsilon$, $0 \leq \delta \leq 1$, and a query $q$, a mechanism $M : \mathcal{X}^n \times \mathcal{Q} \to \mathcal{R}$ will be called $(\epsilon, \delta)$-*differentially-private with respect to $q$* (or *DP*, for short) if for any $s_1, s_1 \in \mathcal{X}^n$ that differ only in one element, the two distributions defined over $\mathcal{R}$ by $M(s_1, q)$ and $M(s_2, q)$ are $(\epsilon, \delta)$-indistinguishable (Definition F.3). In other words, for any $\mathbf{r} \subseteq \mathcal{R}$,

$$
D(\mathbf{r} \,|\, s_1) \leq e^\epsilon \cdot D(\mathbf{r} \,|\, s_2) + \delta,
$$

where the probability is taken over the internal randomness of $M$. Notice that in this definition, there is no probabilistic aspect in the choice of $s$, and the bound is defined on the worst case.

**Definition D.2** (Max Information [DFH+15a]). Given $0 \leq \epsilon$, $0 \leq \delta \leq 1$, a distribution $D_{\mathcal{X}^n}$, and a query $q$, we say a mechanism $M : \mathcal{X}^n \times \mathcal{Q} \to \mathcal{R}$ has *$\delta$-approximate max-information of $\epsilon$ with respect to $D_{\mathcal{X}^n}, q$* (or *MI*, for short) if the two distributions $D_{(\mathcal{X}^n, \mathcal{R})}$ and $D_{\mathcal{X}^n \otimes \mathcal{R}}$ over $\mathcal{X}^n \times \mathcal{R}$ are $(\epsilon, \delta)$-indistinguishable. In other words, for any $\mathbf{b} \subseteq \mathcal{X}^n \times \mathcal{R}$,

$$
D_{(\mathcal{X}^n, \mathcal{R})} (\mathbf{b}) \leq e^\epsilon \cdot D_{\mathcal{X}^n \otimes \mathcal{R}} (\mathbf{b}) + \delta \quad \text{and} \quad D_{\mathcal{X}^n \otimes \mathcal{R}} (\mathbf{b}) \leq e^\epsilon \cdot D_{(\mathcal{X}^n, \mathcal{R})} (\mathbf{b}) + \delta.
$$

Some definitions replace $e$ with 2 as the base of $\epsilon$.

These definition can be extended to apply to a family of queries and/or a family of possible distributions, just like the LSS definition.

**Definition D.3** (Typical Stability, based on Definition 2.3. of [BF16])**.** Given $0 \leq \epsilon$, $0 \leq \delta, \eta \leq 1$, a distribution $D_{\mathcal{X}^n}$, and a query $q$, a mechanism $M : \mathcal{X}^n \times \mathcal{Q} \to \mathcal{R}$ will be called $(\epsilon, \delta, \eta)$-*Typically-Stable with respect to $D_{\mathcal{X}^n}, q$* (or *TS*, for short) if with probability at least $1 - \eta$ over the sampling of $s_1, s_2 \in \mathcal{X}^n$, the conditional distributions induced by the mechanism given the two sets is $(\epsilon, \delta)$-indistinguishable. Formally,

$$\Pr_{S_1, S_2 \sim D_{\mathcal{X}^n}} \left[ \exists \mathbf{r} \subseteq \mathcal{R} \mid D\left(\mathbf{r} \mid S_1\right) > e^\epsilon D\left(\mathbf{r} \mid S_2\right) + \delta \right] < \eta$$

An equivalent definition requires the existence of a subset $\mathbf{s} \in \mathcal{X}^n$, such that (1) $D\left(\mathbf{s}\right) \geq 1 - \eta$, and (2) for any $s_1, s_2 \in \mathbf{s}$

$$D\left(\mathbf{r} \mid s_1\right) \leq e^\epsilon \cdot D\left(\mathbf{r} \mid s_2\right) + \delta$$

Notice that in a way, MI and TS are a natural relaxation of DP, where instead of considering only the probability which is induced by the mechanism, we also consider the underlying distribution.

**Definition D.4** (Bounded Maximal Leakage [EGI19])**.** Given $0 \leq \epsilon$, a distribution $D_{\mathcal{X}^n}$, and a query $q$, a mechanism $M : \mathcal{X}^n \times \mathcal{Q} \to \mathcal{R}$ will be called $\epsilon$-*Bounded-Maximal-Leaking with respect to $D_{\mathcal{X}^n}, q$* (or *ML*, for short) if $\mathcal{L}\left(D_{\mathcal{X}^n} \to D_{\mathcal{R}}\right) \leq \epsilon$, where $\mathcal{L}$ is the Maximal Leakage (Definition F.4).

Similarly to MI, this definition can also be relaxed to the local version.

**Definition D.5** (Bounded Local Maximal Leakage)**.** Given $0 \leq \epsilon$, a distribution $D_{\mathcal{X}^n}$, and a query $q$, a mechanism $M : \mathcal{X}^n \times \mathcal{Q} \to \mathcal{R}$ will be called $\epsilon$-*Bounded-Local-Maximal-Leaking with respect to $D_{\mathcal{X}^n}, q$* (or *ML*, for short) if $\mathcal{L}\left(D_{\mathcal{X}} \to D_{\mathcal{R}}\right) \leq \epsilon$, where $\mathcal{L}$ is the Maximal Leakage (Definition F.4).

In Theorem D.7 we prove that LML implies LMI, the same way ML implies MI.

**Definition D.6** (Compression Scheme [LW86])**.** Given an integer $m < \frac{n}{2}$ and a query $q$, a mechanism $M$ will be said to have a *compression scheme* of size $m$ with respect to $q$ (or *CS*, for short), if $M$ can be described as the composition $f_q \circ g_q$ where the *compression function* $g_q : \mathcal{X}^n \to \mathcal{X}^m$ has the property that $g_q\left(s\right) \subset s$ and $f_q : \mathcal{X}^m \to \mathcal{R}$ is some arbitrary function which will be called the *encoding function*. Both functions might be non deterministic. We will denote $w := g\left(s\right)$ and $r_w := f\left(w\right)$.[14]

One simple case is when $f$ is the identity function, and the mechanism releases $m$ sample elements.

## D.2 Proofs of Implication Theorems

*Proof of Theorem 4.2.* Given $\mathbf{b} \subseteq \mathcal{X} \times \mathcal{R}$ we denote $\mathbf{r_b}\left(x\right) := \left\{r \in \mathcal{R} \mid \left(x, r\right) \in \mathbf{b}\right\}$ (which might be empty for some $x$'s). Using this notation we prove that for any $\mathbf{b} \subseteq \mathcal{X} \times \mathcal{R}$,

$$
\begin{aligned}
D_{(\mathcal{X}, \mathcal{R})}\left(\mathbf{b}\right) &= \sum_{x \in \mathcal{X}} D\left(x\right) D\left(\mathbf{r_b}\left(x\right) \mid x\right) \\
&\stackrel{(1)}{=} \overbrace{\sum_{x' \in \mathcal{X}} D\left(x'\right)}^{=1} \sum_{x \in \mathcal{X}} D\left(x\right) \sum_{s' \in \mathcal{X}^{n-1}} D\left(s'\right) \cdot D\left(\mathbf{r_b}\left(x\right) \mid s' \cup \{x\}\right) \\
&\stackrel{(2)}{\leq} \sum_{x \in \mathcal{X}} D\left(x\right) \sum_{x' \in \mathcal{X}} D\left(x'\right) \sum_{s' \in \mathcal{X}^{n-1}} D\left(s'\right) \left(e^\epsilon \cdot D\left(\mathbf{r_b}\left(x\right) \mid s' \cup \{x'\}\right) + \delta\right) \\
&\stackrel{(1)}{=} \sum_{x \in \mathcal{X}} D\left(x\right) \sum_{s \in \mathcal{X}^n} D\left(s\right) \left(e^\epsilon \cdot D\left(\mathbf{r_b}\left(x\right) \mid s\right) + \delta\right) \\
&= \sum_{x \in \mathcal{X}} D\left(x\right) \left(e^\epsilon \cdot D\left(\mathbf{r_b}\left(x\right)\right) + \delta\right) \\
&= e^\epsilon \cdot D_{\mathcal{X} \otimes \mathcal{R}}\left(\mathbf{b}\right) + \delta,
\end{aligned}
$$

where (1) are a result of the fact that $D_{\mathcal{X}^n}$ is a product distribution, and (2) is a result of the DP definition. The proof is concluded by repeating the same process for the second direction. $\square$

*Proof of Theorem 4.3.* Notice that the proof of DP holding under post-processing (see e.g. [DR$^+$14]), proves in fact that $(\epsilon, \delta)$-indistinguishability is closed under post-processing. Since $x$ is a post-processing of $s$, the fact that $D_{(\mathcal{X}^n, \mathcal{R})}$ and $D_{\mathcal{X}^n \otimes \mathcal{R}}$ are $(\epsilon, \delta)$ indistinguishable implies that $D_{(\mathcal{X}, \mathcal{R})}$ and $D_{\mathcal{X} \otimes \mathcal{R}}$ are indistinguishable as well. $\qquad\square$

*Proof of Theorem 4.4.* Given any $\mathbf{b} \subseteq \mathcal{X} \times \mathcal{R}$ we denote $\mathbf{r_b}(x) := \{r \in \mathcal{R} \mid (x, r) \in \mathbf{b}\}$ (which might be empty for some $x$'s). Using this notation and the subset $\mathbf{s}$ from Definition D.3 we prove that for any $\mathbf{b} \subseteq \mathcal{X} \times \mathcal{R}$,

$$D_{(\mathcal{X}, \mathcal{R})}(\mathbf{b}) = \sum_{x \in \mathcal{X}} D(x) D(\mathbf{r_b}(x) \mid x)$$

$$\stackrel{(1)}{=} \sum_{x \in \mathcal{X}} \sum_{s \in \mathcal{X}^n} D(s) D(x \mid s) \overbrace{\sum_{s' \in \mathcal{X}^n} D(s') D(\mathbf{r_b}(x) \mid s)}^{=1}$$

$$\stackrel{(2)}{\leq} e^\epsilon \sum_{x \in \mathcal{X}} \sum_{s \in \mathbf{s}} D(s) D(x \mid s) \sum_{s' \in \mathbf{s}} D(s') D(\mathbf{r_b}(x) \mid s') + \delta + 2\eta$$

$$\leq e^\epsilon \sum_{x \in \mathcal{X}} \overbrace{\sum_{s \in \mathcal{X}^n} D(s) D(x \mid s)}^{=D(x)} \overbrace{\sum_{s' \in \mathcal{X}^n} D(s') D(\mathbf{r_b}(x) \mid s')}^{=D(\mathbf{r_b}(x))} + \delta + 2\eta$$

$$= e^\epsilon D_{\mathcal{X} \otimes \mathcal{R}}(\mathbf{b}) + \delta + 2\eta$$

where (1) results from the fact that $x$ and $r$ are independent given $s$, and (2) from the definition of TS. $\qquad\square$

*Proof of Theorem 4.5.* Assume $M$ is not $\left(\epsilon', \frac{\delta}{\epsilon}\right)$-LSS, which means that in particular $D(\mathbf{r}_{\epsilon'}) > \frac{\delta}{\epsilon}$. Denoting $B^{\epsilon'}_{\mathcal{X} \times \mathcal{R}} := \bigcup_{r \in \mathbf{r}_{\epsilon'}} (\mathbf{x}_+(r) \times \{r\})$ we get that from the definition of the stability loss,

$$D_{(\mathcal{X}, \mathcal{R})}\left(B^{\epsilon'}_{\mathcal{X} \times \mathcal{R}}\right) - D_{\mathcal{X} \otimes \mathcal{R}}\left(B^{\epsilon'}_{\mathcal{X} \times \mathcal{R}}\right) = \sum_{r \in \mathbf{r}_{\epsilon'}} D(r) \cdot \ell(r) > \epsilon' \cdot D(\mathbf{r}_{\epsilon'}).$$

But on the other hand, from the fact that $M$ is $(\epsilon, \delta)$-LMI we get in contradiction that

$$D_{(\mathcal{X}, \mathcal{R})}\left(B^{\epsilon' \epsilon'}_{\mathcal{X} \times \mathcal{R}}\right) - D_{\mathcal{X} \otimes \mathcal{R}}\left(B^{\epsilon'}_{\mathcal{X} \times \mathcal{R}}\right) \leq D_{\mathcal{X} \otimes \mathcal{R}}\left(B^{\epsilon'}_{\mathcal{X} \times \mathcal{R}}\right) \cdot (e^\epsilon - 1) + \delta$$

$$\stackrel{(1)}{\leq} D(\mathbf{r}_{\epsilon'}) \cdot (e^\epsilon - 1) + \epsilon \cdot D(\mathbf{r}_{\epsilon'})$$

$$\stackrel{(2)}{\leq} \epsilon' \cdot D(\mathbf{r}_{\epsilon'})$$

where (1) results from the fact that $\epsilon \cdot D(\mathbf{r}_{\epsilon'}) > \delta$, and (2) from the definition of $\epsilon'$ and the assumption that $M$ is not $\left(\epsilon', \frac{\delta}{\epsilon}\right)$-LSS. The proof is concluded by repeating the same process for the second direction. $\qquad\square$

**Theorem D.7** (Local Bounded Maximal Leakage implies Local Max Information)**.** *Given* $0 \leq \epsilon$, $0 < \delta \leq 1$ *a distribution* $D_{\mathcal{X}^n}$ *and a query* $q$, *if a mechanism* $M$ *is* $\epsilon$-LML *with respect to* $D_{\mathcal{X}^n}$ *and* $q$, *then it is* $\left(\epsilon + \ln\left(\frac{1}{\delta}\right), \delta\right)$-LMI *with respect to the same* $D_{\mathcal{X}^n}$ *and* $q$.

*Proof.* The proof is identical to the one used by [EGI19] when proving that ML implies MI (Theorem 7). $\qquad\square$

**Lemma D.8** (see, e.g., [SSBD14] Theorem 30.2)**.** *Given* $0 \leq \delta \leq 1$, $m \leq \frac{n}{2}$, *a domain* $\mathcal{X}$, *and a distribution* $D_{\mathcal{X}}$ *defined over it, we denote by* $\mathcal{H}$ *the family of functions (usually referred to as* hypothesis *in the context of Machine Learning) of the form* $h : \mathcal{X} \to \{0, 1\}$, *and let* $h^* \in \mathcal{H}$ *be some unique hypothesis which we will think of as the* true *hypothesis. We will refer to* $h^*(x)$ *as the true* label *of* $x$, *and denote the labeled domain by* $\mathcal{X}_{h^*} := \{(x, h^*(x)) \mid x \in \mathcal{X}\}$. *Let* $M : \mathcal{X}^n \times \mathcal{Q} \to \mathcal{H}$ *be a mechanism with a compression scheme (Definition D.6), In this case, with probability (over the*

*sampling of s and the internal randomness of the mechanism in case it is non deterministic) greater then* $1 - \delta$ *we have that,*

$$|h_w (s \setminus w) - h_w (D_{\mathcal{X}})| \leq \sqrt{h_w (s \setminus w) \frac{4m\ln (2n/\delta)}{n}} + \frac{8m\ln (2n/\delta)}{n}$$

*where* $h_w (s \setminus w)$ *is the empirical mean of* $h_w$ *over* $s \setminus w$ *and* $h_w (D_{\mathcal{X}})$ *is its expectation with respect to* $D_{\mathcal{X}}$.

*Proof of Theorem 4.6.* We will prove that $g$ is $(\epsilon, \delta)$-LSS for such an $\epsilon$, and since LSS holds under post-processing, this suffices. Notice that now $\mathcal{R} = \mathcal{X}^m$. This proof resembles that of [CLN$^+$16].

We start by analyzing the loss of $w$ and get that,

$$
\begin{aligned}
\ell (w) &= \sum_{x \in \mathbf{x}_+(w)} (D (x \,|\, w) - D (x)) \\
&= \sum_{x \in \mathbf{x}_+(w)} \sum_{s \in \mathcal{X}^n} D (s \,|\, w) (D (x \,|\, s) - D (x)) \\
&= \sum_{s \in \mathcal{X}^n} D (s \,|\, w) \sum_{x \in \mathbf{x}_+(w)} \left( \frac{m}{n} D (x \,|\, w) + \frac{n-m}{n} D (x \,|\, s \setminus w) - D (x) \right) \\
&\leq \sum_{s \in \mathcal{X}^n} D (s \,|\, w) \left( \frac{m}{n} + \sum_{x \in \mathbf{x}_+(w)} (D (x \,|\, s \setminus w) - D (x)) \right) \\
&= \sum_{s \in \mathcal{X}^n} D (s \,|\, w) \left( \frac{m}{n} + \sum_{x \in \mathcal{X}} (D (x \,|\, s \setminus w) - D (x)) h_w^+ (x) \right) \\
&= \sum_{s \in \mathcal{X}^n} D (s \,|\, w) \left( \frac{m}{n} + h_w^+ (s \setminus w) - h_w^+ (D_{\mathcal{X}}) \right)
\end{aligned}
$$

where $h_w^+ (x)$ is simply the characteristic function of $\mathbf{x}_+ (w)$.

Using this inequality we get that $\forall \mathbf{r} \subseteq \mathcal{R}$,

$$
\begin{aligned}
D (\mathbf{r}) (\ell (\mathbf{r}) - \epsilon) &= \sum_{w \in \mathbf{r}} D (w) (\ell (w) - \epsilon) \\
&\overset{(1)}{\leq} \sum_{w \in \mathbf{r}} D (w) \sum_{s \in \mathcal{X}^n} D (s \,|\, w) \left( \frac{m}{n} + h_w^+ (s \setminus w) - h_w^+ (D_{\mathcal{X}}) - \epsilon \right) \\
&= \sum_{s \in \mathcal{X}^n} D (s) \sum_{w \in \mathbf{r}} D (w \,|\, s) \left( h_w^+ (s \setminus w) - h_w^+ (D_{\mathcal{X}}) - \left( \epsilon - \frac{m}{n} \right) \right) \\
&\leq \sum_{s \in \mathcal{X}^n} D (s) \max_{w=g(s), h=f(w)} \left( h (s \setminus w) - h (D_{\mathcal{X}}) - \left( \epsilon - \frac{m}{n} \right) \right) \\
&\overset{(2)}{\leq} \Pr_{S \sim D_{\mathcal{X}^n}, W \sim g(S), H \sim f(W)} \left[ H (S \setminus W) - H (D_{\mathcal{X}}) > \left( \epsilon - \frac{m}{n} \right) \right] \\
&\overset{(3)}{\leq} \sqrt{\frac{4m\ln (2n/\delta)}{n}} + \frac{8m\ln (2n/\delta)}{n} + \frac{m}{n} \\
&\overset{(4)}{\leq} 11 \sqrt{\frac{m\ln (2n/\delta)}{n}}
\end{aligned}
$$

where (1) results from the previous inequality, (2) from the fact that we removed $s$'s for which the summand is negative, and replaced the positive ones with 1 - which is greater then the maximal possible value, (3) from Lemma D.8 and the fact that the value of $h$ is bounded by 1, and (4) from the fact that $m \leq \frac{n}{9\ln\left(\frac{2n}{\delta}\right)}$. $\qquad\square$

### D.3 Proofs of Separation Theorems

*Proof of Theorem 4.7.* Without loss of generality, assume $0 < \epsilon \leq 0.7$. Given $0 \leq \alpha \leq \frac{\epsilon}{7}, p = \frac{1}{2} + \alpha$ we will define some function $f : \mathcal{X} \to \{0, 1\}$, and for $i \in \{0, 1\}$ denote $\mathbf{x}_i := \{x \in \mathcal{X} \mid f(x) = i\}$, set an arbitrary distribution $D_{\mathcal{X}}$ such that $D(\mathbf{x}_1) = p$, and $D_{\mathcal{X}^n}$ which is the product of $D_{\mathcal{X}}$. We will consider a mechanism $M$ which in response to a query $q$ returns the parity function of the vector $(f(s_1), \ldots, f(s_n))$, where $s_1, \ldots s_n$ denotes the elements of the sample set $s$. Formally, $M(q, s) = |s \cap \mathbf{x}_1| \pmod 2$, and we prove that this mechanism is $(\epsilon, 0)$-LMI but not $\left(1, \frac{1}{5}\right)$-MI.

We start with denoting by $p_{n-2}$ the probability that the parity function of a sample of size $n - 2$ will be equal to 1, and the possible outputs as $r_0, r_1$. Notice that,

$$D(r_1 \mid \mathbf{x}_1) = p \cdot p_{n-2} + (1 - p)(1 - p_{n-2})$$
$$D(r_1 \mid \mathbf{x}_0) = (1 - p) p_{n-2} + p(1 - p_{n-2}) = 1 - D(r_1 \mid \mathbf{x}_1)$$
$$D(r_1) = p \cdot D(r_1 \mid \mathbf{x}_1) + (1 - p) D(r_1 \mid \mathbf{x}_0)$$
$$= (2p - 1) D(r_1 \mid \mathbf{x}_1) + 1 - p$$
$$= (1 - 2p) D(r_1 \mid \mathbf{x}_0) + p$$

Using these identities we will first prove that $\frac{D(r_1)}{D(r_1 \mid \mathbf{x}_1)}, \frac{D(r_1)}{D(r_1 \mid \mathbf{x}_0)} \leq e^{\epsilon}$. Since a similar claim can be proven for $\frac{D(r_1 \mid \mathbf{x}_1)}{D(r_1)}, \frac{D(r_1 \mid \mathbf{x}_0)}{D(r_1)}$, we get that this mechanism is $(\epsilon, 0)$-LMI.

$$\frac{D(r_1)}{D(r_1 \mid \mathbf{x}_1)} = \frac{(2p - 1) D(r_1 \mid \mathbf{x}_1) + 1 - p}{D(r_1 \mid \mathbf{x}_1)}$$
$$= 2p - 1 + \frac{1 - p}{(2p - 1) p_{n-2} + 1 - p}$$
$$= 2p - \frac{(2p - 1) p_{n-2}}{(2p - 1) p_{n-2} + 1 - p}$$
$$= 1 + 2\alpha - \overbrace{\frac{2\alpha p_{n-2}}{\alpha(2p_{n-2} - 1) + \frac{1}{2}}}^{\geq 0}$$
$$\overset{(1)}{\leq} 1 + \overbrace{2\alpha}^{\leq \epsilon}$$
$$\overset{(2)}{\leq} e^{\epsilon}$$

where (1) results from the fact that $0 \leq \alpha < \frac{\epsilon}{7} \leq \frac{1}{10}$, so the denominator $\alpha(2p_{n-2} - 1) + \frac{1}{2}$ must be positive, and (2) is a result of the inequality $1 + \epsilon \leq e^{\epsilon}$ for any $\epsilon < 1$. Similarly we get that,

$$\frac{D(r_1)}{D(r_1 \mid \mathbf{x}_0)} = \frac{(1 - 2p) D(r_1 \mid \mathbf{x}_0) + p}{D(r_1 \mid \mathbf{x}_0)}$$
$$= 1 - 2p + \frac{p}{D(r_1 \mid \mathbf{x}_0)}$$
$$= 2 - 2p - \frac{(1 - 2p) p_{n-2}}{(1 - 2p) p_{n-2} + p}$$
$$= 1 + 2\alpha + \overbrace{\frac{2\alpha \cdot p_{n-2}}{\alpha(1 - 2p_{n-2}) + \frac{1}{2}}}^{\leq 5\alpha}$$
$$\overset{(1)}{\leq} 1 + \overbrace{7\alpha}^{\leq \epsilon}$$
$$\overset{(2)}{\leq} e^{\epsilon}$$

where (1) results from the fact that $0 \leq \alpha < \frac{\epsilon}{7} \leq \frac{1}{10}$, and $0 \leq p_{n-2} \leq 1$, so $\alpha \left(1 - 2p_{n-2}\right) + \frac{1}{2} \geq \frac{4}{10}$, and (2) is a result of the inequality $1 + \epsilon \leq e^{\epsilon}$ for any $\epsilon < 1$.

On the other hand, we will prove the response dramatically changes the distribution over the sample sets. Using the fact that the parity function of a Binomial random variable $\mathbf{b}(n, p)$ is a Bernoulli random variable $\mathrm{Ber}\left(\frac{1 - (1-2p)^n}{2}\right)$, and denoting $\mathcal{S}_1$ the set of all sample sets with parity value 1, we get that,

$$
\begin{aligned}
D_{\mathcal{X}^n \otimes \mathcal{R}}\left(\mathcal{S}_1 \times \{r_0\}\right) &= \overbrace{D\left(\mathcal{S}_1\right)}^{D(r_1)} \cdot D\left(r_0\right) \\
&= \frac{1 - (1-2p)^{2n}}{4} \\
&= e^1 \overbrace{D_{(\mathcal{X}^n, \mathcal{R})}\left(\mathcal{S}_1 \times \{r_0\}\right)}^{=0} + \frac{1 - (2\alpha)^{2n}}{4} \\
&\overset{(1)}{>} e^1 D_{(\mathcal{X}^n, \mathcal{R})}\left(\mathcal{S}_1 \times \{r_0\}\right) + \frac{1}{5}
\end{aligned}
$$

where (1) is a result of the fact that $0 \leq \alpha < \frac{\epsilon}{7} \leq \frac{1}{10}$, $n \geq 3$ and $\frac{1 - \left(\frac{1}{5}\right)^6}{4} > \frac{1}{5}$, which means this mechanism is not $\left(1, \frac{1}{5}\right)$-MI.

$\square$

*Proof of Theorem 4.8.* Without loss of generality $0 \leq \delta \leq 0.1$, so $n > 2\ln\left(\frac{2}{\delta}\right)$. Given $N > n^2$, $\mathcal{X} := [N]$, an arbitrary $D_{\mathcal{X}}$ such that $\forall x \in \mathcal{X} : D_{\mathcal{X}}(x) \leq \frac{1}{n^2}$, and $D_{\mathcal{X}^n}$ which is the product of $D_{\mathcal{X}}$, we consider a mechanism $M$ which in response to some query $q$ uniformly samples one element from its sample set and outputs it.

The fact that this mechanism is $\left(11\sqrt{\frac{\ln(2n/\delta)}{n}}, \delta\right)$-LSS is a direct result of Theorem 4.6 for $m = 1$.

On the other hand, notice that any $r \in \mathcal{R}$ encodes one sample element which we will denote by $x(r)$. Using this notation we will define the set $\mathbf{b} := \underset{r \in \mathcal{R}}{\cup}(x(r), r)$.

$$
\begin{aligned}
D_{(\mathcal{X}, \mathcal{R})}\left(\mathbf{b}\right) &= \sum_{r \in \mathcal{R}} D(r) \cdot D\left(x(r) \mid r\right) \\
&\overset{(1)}{\geq} \sum_{r \in \mathcal{R}} D(r) \cdot \frac{1}{n} \\
&\overset{(2)}{>} \sum_{r \in \mathcal{R}} D(r) e \frac{1}{n^2} + \overbrace{\sum_{r \in \mathcal{R}} D(r)}^{=1} \frac{1}{2n} \\
&\geq e \sum_{r \in \mathcal{R}} D(r) \cdot \overbrace{D\left(x(r)\right)}^{\leq \frac{1}{n^2}} + \frac{1}{2n} \\
&= e^1 \cdot D_{\mathcal{X} \otimes \mathcal{R}}\left(\mathbf{b}\right) + \frac{1}{2n}
\end{aligned}
$$

where (1) is a result of the fact that if all elements in the sample set differ from each other, with probability $\frac{1}{n}$ the sampling mechanism will return the same sample element which was encoded by $r$ and if not then the probability is only higher, and (2) is a result of the definitions of $\delta$ and $n$. This proves the mechanism is not $\left(1, \frac{1}{2n}\right)$-LMI.

$\square$

## E   Missing Details from Section 5

Definitions and properties in this section are due to [DR$^+$14].

**Definition E.1** (Laplace Mechanism). Given $0 \le b$ and a query $q \in \mathcal{Q}_\Delta$, the Laplace mechanism with parameter $b$ is defined as:
$$M(s, q) = q(s) + \text{Lap}_b$$
where $\text{Lap}_b$ is a random variable with unbiased Laplace distribution, which if a symmetric exponential distribution. Formally:
$$\text{Lap}_b(x) = \frac{1}{2b} e^{-\frac{|x|}{b}}$$

**Theorem E.2** (Laplace Mechanism is Differentially Private). *Given $0 \le b, \epsilon$ and a query $q \in \mathcal{Q}_\Delta$, the Laplace mechanism with parameter $b$ is $\left( \frac{2\Delta}{n \cdot b}, 0 \right)$-DP.*

**Theorem E.3** (Laplace Mechanism is Sample Accurate). *Given $0 \le b$, $0 < \delta \le 1$ and a query $q \in \mathcal{Q}_\Delta$, the Laplace mechanism with parameter $b$ is $\left( b \cdot \ln\left(\frac{1}{\delta}\right), \delta \right)$-Sample Accurate.*

**Definition E.4** (Gaussian Mechanism). Given $0 \le \sigma$ and a query $q \in \mathcal{Q}_\Delta$, the Gaussian mechanism with parameter $\sigma$ is defined as:
$$M(s, q) = q(s) + \text{G}_\sigma$$
where $\text{G}_\sigma$ is a random variable with unbiased Gaussian distribution and standard deviation $\sigma$.

**Theorem E.5** (Gaussian Mechanism is Differentially Private). *Given $0 \le \sigma, \epsilon$, $0 < \delta \le 1$, and a query $q \in \mathcal{Q}_\Delta$, the Gaussian mechanism with parameter $\sigma$ is $\left( \frac{2\Delta\sqrt{2\ln(1.25/\delta)}}{n\sigma}, \delta \right)$-DP.*

**Theorem E.6** (Gaussian Mechanism is Sample Accurate). *Given $0 \le \sigma, \epsilon$, $0 < \delta \le 1$, and a query $q \in \mathcal{Q}_\Delta$, the Gaussian mechanism with parameter $\sigma$ is $\left( \frac{\epsilon}{\sqrt{2\ln\left(\sqrt{2}/\pi\delta\right)}}, \delta \right)$-Sample Accurate.*

# F   Distance Measures on Distributions

These distance measures between distributions will be used in various places in the paper.

**Definition F.1** (Statistical Distance). The *Statistical Distance* (also know as *Total Variation Distance*) between two probability distributions $D_1, D_2$ over some domain $\mathcal{R}$ is defined as,

$$
\begin{aligned}
\text{SD}(D_1, D_2) &:= \max_{\mathbf{r} \in \mathcal{R}} (D_1(\mathbf{r}) - D_2(\mathbf{r})) \\
&= \max_{\mathbf{r} \in \mathcal{R}} (D_2(\mathbf{r}) - D_1(\mathbf{r})) \\
&= \frac{1}{2} \cdot \sum_{r \in \mathcal{R}} |D_1(r) - D_2(r)|.
\end{aligned}
$$

The maximal set in the first definition is simply the set of all $r$'s for which $D_1(r) > D_2(r)$ and for the second - the set of all $r$'s for which $D_1(r) < D_2(r)$

**Definition F.2** ($\delta$-approximate max divergence). The *$\delta$-approximate max divergence* between two probability distributions $D_1, D_2$ over some domain $\mathcal{R}$ is defined as

$$\mathbf{D}_\infty^\delta(D_1 \| D_2) := \max_{\mathbf{r} \subseteq \text{Supp}(D_1) \wedge D_1(\mathbf{r}) \ge \delta} \ln\left( \frac{D_1(\mathbf{r}) - \delta}{D_2(\mathbf{r})} \right).$$

The case where $\delta = 0$ is simply called the *max divergence*.

**Definition F.3** (Indistinguishable distributions). Two probability distributions $D_1, D_2$ over some domain $\mathcal{R}$ will be called *$(\epsilon, \delta)$-indistinguishable* if

$$\max \left\{ \mathbf{D}_\infty^\delta(D_1 \| D_2), \mathbf{D}_\infty^\delta(D_2 \| D_1) \right\} \le \epsilon.$$

this can also be written as the condition that for any $\mathbf{r} \subseteq \mathcal{R}$
$$D_1(\mathbf{r}) \le e^\epsilon \cdot D_2(\mathbf{r}) + \delta \quad \text{and} \quad D_2(\mathbf{r}) \le e^\epsilon \cdot D_1(\mathbf{r}) + \delta$$

**Definition F.4** (Maximal Leakage, based on [IWK18]). Given two finite domains $\mathcal{X}, \mathcal{Y}$ and a joint distribution $D_{(\mathcal{X}, \mathcal{Y})}$ defined over $\mathcal{X} \times \mathcal{Y}$, The *Maximal Leakage* between two marginal distributions $D_\mathcal{X}, D_\mathcal{Y}$ is defined as,

$$\mathcal{L}(D_\mathcal{X} \to D_\mathcal{Y}) := \log\left( \sum_{y \in \mathcal{Y}} \max_{x \in \mathcal{X} \mid D(x) > 0} D(y \mid x) \right).$$