[Reviews · NeurIPS 2019]

Reviewer 1



I like the paper, and support to accept it. The major contribution is that the authors define a new measure of stability, which is both necessary and sufficient for generalization. Understanding specialized necessary and sufficient properties for restricted classes of queries and distributions is an interesting direction for future work. Lifting the condition that the mechanisms need to be sample accurate is not covered in the current formalization, and is an important direction of further research. Writing style: The paper introduces a lot of notation and jargon, and I found it to be very hard to read. Simplifying the paper in the final version will definitely increase the accessibility and is strongly recommended. Post feedback: Thanks for the feedback. I am happy that authors would invest time in improving the writing style of the paper. I still support to accept it.

Reviewer 2



This paper contributes to the line of works aiming at identifying stability notions that can be used to guarantee generalization in adaptive data analysis. The notion that the paper proposes is based on a measure of stability loss defined in terms of statistical distance between the prior distribution over the possible inputs and the posterior distribution over the inputs induced by an output of the mechanism. The way the paper bounds this notion is a reminiscence of the epsilon,delta loss bounds. This notions seems to have the good properties that one would like in notions to guarantee generalization and it also relates well with the notions that have been previously studied in this line of work. One thing that is missing in the paper is a discussion of how to enforce a bound on this measure and so have good generalization properties, also just a discussion of how this quantity can be estimated is missing. This makes quite difficult to understand the effective usefulness of this notion. Another minor issue I have with this paper is in the way the separation results are formulated. Perhaps I am being too formal but Theorem 4.6 and Theorem 4.7 seem very specific to particular values of LMI. Could this be formulated in a more general way for arbitrary values of the parameters? It would be great if the authors can clarify this aspect in the rebuttal. Besides these two issues I find this an interesting and well written paper. Pros: -novel notion that guarantees adaptive generalization -the paper shows that previous notions guaranteeing generalization all imply a bound on Local Statistical Stability Cons: -unclear the scope of the separation results -the paper doesn't provide any method to guarantee a bound on the Local Statistical Stability, neither any concrete example which satisfies it. Comments after rebuttal ---------------------------- Thanks for your answers and for clarifying some of the doubts I had on this paper. Please, include a more thorough discussion on possible directions to find methods to enforce LSS in future versions of the paper. Please, also add a discussion about the relations between LSS and other stability notions guaranteeing generalization.

Reviewer 3



What I like about the paper is that is provides a unifying theory for various stability notions that have been discussed in the literature. One common thread that existing algorithms were known to have is post-hoc generalization, but it was later proved that this notion does not compose adaptively, meaning that, if individual algorithms prevent overfitting in the sense of post-hoc generalization, combining them may permit overfitting. The current notion, LSS, composes adaptively, similarly to differential privacy (DP). However, LSS is proved to be weaker than DP, but still sufficient to guarantee generalization. This proof is given via the monitor argument of Bassily et al. Under no further modeling of the general problem of adaptive data analysis, from previous results in the area we know that sqrt(k)/epsilon^2 samples are sufficient and necessary for generalization, where k is the number of queries and epsilon is the target accuracy, and we know differential privacy can guarantee this complexity, so I am not sure about direct practical implications of this work. For now I see the advantages as being primarily of theoretical nature, however the possible directions given in Section 5 sound promising. In Definition 2.1., is there a specific reason why total variation distance is used? Could similar results have been proved with a different distance between distributions? I’m not sure how to reason about the alpha_i sequence in Theorem 2.7 (this has to interplay somehow with the sample accuracy requirement, and without an explicit mechanism it’s hard to analyze this trade-off). That aside, I don’t completely agree with the comment made just below, that the theorem is non-trivial for alpha_i <= epsilon_i. In particular, if alpha_i = epsilon_i, then the result is strictly worse than that of Theorem 2.7? The main weakness of this paper is that it is not clear what the space of LSS mechanisms is. It possibly allows mechanisms that give us all the benefits of DP, but circumvent other issues (or under some modeling assumptions they might even outperform DP). But currently the definition is too general to see explicit mechanisms. (Since the mechanisms don’t know the prior data distribution, it’s hard to see how to guarantee LSS.) 
I wonder if this setup can be used to say something when the prior of the adversary is off. Currently it’s assumed that the prior is the actual data distribution, but does this framework allow saying something about overfitting if we know how much the adversary’s prior deviates from the true one? Minor comment: Is there a reason why sequences in X^n (possibly non-iid samples) are considered? I didn’t see any motivation for non-iid data sets, however considering a distribution over sequences seemed to make notation and discussion unnecessarily more complicated at times. Overall I think the paper gives some interesting theoretical insights. It’s hard to see at this point what its practical implications might be. Post-feedback: I thank the authors for clarifying some confusions that I had, and I'm happy to see the authors are also thinking of novel mechanisms, as suggested in the reviews. I think this paper makes a good theoretical contribution and I still recommend acceptance.

[Author Response · NeurIPS 2019]

Thank you to all of the reviewers for their insightful comments! We respond to specific questions and comments of each
reviewer below, and further provide additional discussion of the problem of developing algorithms that guarantee LSS.

**Reviewer 1:** We completely agree that the notation and style should be streamlined to improve readability, and have
undertaken changes to address this. One such change will be removing subscripts and superscripts in cases where the
distribution/generator/etc. are clear from context.

**Reviewer 2:** Indeed, the way Theorems 4.6. and 4.7 were stated may have made them seem weaker then they are
actually are, and we thank you for pointing this out. The theorems were stated for the "interesting" values of small $\epsilon$
and $\delta$, but also immediately hold for larger values, so long as the sample size is sufficient. Theorem 4.6 holds for *any*
$\epsilon > 0$, for sufficiently large $n$, and Theorem 4.7 similarly holds for *any* $0 < \delta < 1$, for sufficiently large $n$. We have
revised these two Theorem statements to reflect this.

**Reviewers 2 and 3:** We agree with you that developing algorithms that satisfy LSS and techniques for bounding LSS
is an exciting direction for future work, and we would be glad to expand our discussion of this in the paper.

Naturally, any mechanism which guarantees Differential privacy (e.g., the Laplace and Gaussian mechanisms) will
guarantee LSS as well, as a result of the DP->LMI->LSS implications. We plan to point this out more explicitly.

One can also see this, and perhaps gain additional insight, by manipulating the loss definition:

$$\sum_{x \in X_+(r)} \left( D_{\mathcal{X}|\mathcal{R}}^G \left( x \,|\, r \right) - D_{\mathcal{X}} \left( x \right) \right) = \sum_{x \in X_+(r)} D_{\mathcal{X}} \left( x \right) \left( \frac{D_{\mathcal{X}|\mathcal{R}}^G \left( x \,|\, r \right)}{D_{\mathcal{X}} \left( x \right)} - 1 \right) = \sum_{x \in X_+(r)} D_{\mathcal{X}} \left( x \right) \left( \frac{D_{\mathcal{R}|\mathcal{X}}^G \left( r \,|\, x \right)}{D_{\mathcal{R}}^G \left( r \right)} - 1 \right)$$

Since $D_{\mathcal{R}}^G \left( r \right) = \sum_{x' \in \mathcal{X}} D_{\mathcal{R}|\mathcal{X}}^G \left( r \,|\, x' \right)$, it suffices to bound the quantity $\frac{D_{\mathcal{R}|\mathcal{X}}^G \left( r \,|\, x \right)}{D_{\mathcal{R}|\mathcal{X}}^G \left( r \,|\, x' \right)}$ for any $x, x' \in \mathcal{X}$, which is
bounded by $e^\epsilon$ for a Laplace mechanism with parameter $\frac{\Delta}{n\epsilon}$ in the case of a product distribution. Though this example
does not provide direct improvement over DP, it may suggest a potential technique for proving LSS bounds for novel
mechanisms.

**Reviewer 3:** There are indeed other candidates for the distance notion in Definition 2.1. We have explored some of
them, but have not found another notion that we can show is both necessary and sufficient for generalization. Perhaps
the most natural alternative to consider is bounded KL-Divergence, which, by Jensen's inequality, implies a bound on
TV-distance. Thus, it is natural that bounded KL-Divergence would be sufficient for generalization; however, it is not
clear that it is necessary. The form of the "loss assessment query" we introduce provides some intuition for the choice
of the TV-distance; one cannot construct a natural analogous query for KL-Divergence, due to its unboundedness. This
observation does not demonstrate that other distance measures cannot be used, but at least suggests that our proof
technique may not suit them.

The fact that we handle non-iid databases is actually crucial. The reason for this is that even if the underlying data
distribution were iid, the resulting *posterior* distribution given a query response might no longer be iid. Thanks for
pointing out that we need to clarify this in the writeup.

The $\alpha_i$ values presented in Theorem 2.7 are expected losses, which might be significantly lower than $\epsilon_i$ (which can be
thought of as high probability bounds on the loss). As you suggest, we will clarify the comment right after Theorem
2.7, that the Theorem is weakest when $\alpha_i$ is close to $\epsilon_i$, and more meaningful when $\alpha_i \ll \epsilon_i$.

[Meta-Review · NeurIPS 2019]

The paper introduces a new notion of stability (LSS) and shows formally that it's sufficient and necessary for generalization in adaptive settings. The necessity result is particularly interesting. The paper also gives a rigorous comparison with the notions of approximate max-information and differential privacy. The paper makes a good, clear theoretical contribution to the area of adaptive data analysis. The paper would have been much stronger if it gives (i) new non-trivial mechanisms achieving LSS, and (ii) extended comparisons with other notions such as typical stability and maximal leakage. As recommended in the post-rebuttal discussion, the authors are encouraged to provide at least some discussion on the possible directions for developing non-trivial LSS mechanisms and what general properties they would expect these mechanisms to have. It is also recommended that the authors make a more comprehensive comparison with other existing notions of stability that guarantee generalization in adaptive data analysis.